# Provable Benefits of Actor-Critic Methods for Offline Reinforcement Learning

**Andrea Zanette**[*]
University of California, Berkeley
zanette@berkeley.edu

**Martin J. Wainwright**
University of California, Berkeley
wainwrig@berkeley.edu

**Emma Brunskill**
Stanford University
ebrun@stanford.edu

## Abstract

Actor-critic methods are widely used in offline reinforcement learning practice, but are not so well-understood theoretically. We propose a new offline actor-critic algorithm that naturally incorporates the pessimism principle, leading to several key advantages compared to the state of the art. The algorithm can operate when the Bellman evaluation operator is closed with respect to the action value function of the actor's policies; this is a more general setting than the low-rank MDP model. Despite the added generality, the procedure is computationally tractable as it involves the solution of a sequence of second-order programs. We prove an upper bound on the suboptimality gap of the policy returned by the procedure that depends on the data coverage of any arbitrary, possibly data dependent comparator policy. The achievable guarantee is complemented with a minimax lower bound that is matching up to logarithmic factors.

## 1 Introduction

The problem of learning a near-optimal policy is a core challenge in reinforcement learning (RL). In many settings, it is beneficial to be able to learn a good policy using only a pre-collected set of data, without further exploration with the environment; this problem is known as *offline or batch policy learning*. The offline setting has unique challenges due to the incomplete information about the Markov decision process (MDP) encoded in the available dataset. For example, due to maximization bias, a naive offline algorithm can return a policy with a severely overestimated value. In order to avoid such undesirable behavior, researchers have introduced the idea of pessimism under uncertainty, and there is now a growing literature (e.g., Liu et al. (2020); Jin et al. (2020b); Buckman et al. (2020); Kumar et al. (2019); Kidambi et al. (2020); Yu et al. (2020)) on different ways in which pessimism can be incorporated. See Appendix B for additional references and discussion of this body of work.

At a high level, incorporating pessimism prevents algorithms from settling down on uncertain policies whose value might be misleadingly high under the current dataset due to statistical errors. By using pessimism, uncertain policies are penalized in such a way that only those policies robust to statistical errors are returned. The principle can be implemented in at least two different ways: (a) by penalizing policies that are far from the one that generated the dataset; or (b) by penalizing the value functions of policies not well covered by the dataset. In this paper, we take the latter avenue.

---

[*]This work was fully completed while Andrea Zanette was a PhD candidate at Stanford University. Future updates of this work will be available at `https://arxiv.org/abs/2108.08812`

35th Conference on Neural Information Processing Systems (NeurIPS 2021).

## 1.1 Overview and our contributions

Implementing pessimism with function approximation is challenging for several reasons. First, uncertainty must be estimated with particular care. On one hand, underestimating it can fail to correct the coverage problem. On the other hand, overestimating it leads to policies that are too conservative and thus underperform. Second, the incorporation of pessimism may introduce complex, higher order perturbations into the value function class handled by the algorithm. Similar issues can arise when adding optimistic bonuses in the exploration. The increased complexity of the function class often requires additional assumptions on the model, because the new class needs to interact "nicely" with the Bellman operator. Prior art on pessimism with function approximation has by-passed this problem by making strong model assumptions, such as low-rank transitions Jin et al. (2020b) or algorithm-specific assumptions Liu et al. (2020).

**Actor-critic methods:** Most past theoretical work on offline reinforcement learning on finding with high probability the policy with the highest performance has focused on algorithms that are either model or value-based[2] Liu et al. (2020); Jin et al. (2020b); Buckman et al. (2020); Kidambi et al. (2020); Yu et al. (2020); these often incorporate pessimism into the estimates of the policy performance. Actor-critic methods are a hybrid class of methods that mitigate some deficiencies of methods that are either purely policy or purely value-based Konda and Tsitsiklis (2000, 2003); Heess et al. (2015); Haarnoja et al. (2017, 2018); in modern RL, they are widely used in practice (e.g., Levine et al. (2020); Wu et al. (2019, 2021); Kumar et al. (2019, 2020)). An actor-critic method generally consists of an actor that changes the policy in order to maximize its value as estimated by the critic. Given their popularity, it is natural to ask the following question: *do actor-critic methods provably offer any advantage in offline RL?* The main contribution of this paper is to give a positive answer to this question: by separating the policy optimization from the policy evaluation, both tasks become simpler to design and the pessimism principle can be incorporated more naturally.

**Contributions:** More specifically, we study the problem of policy learning using linear function approximation in the offline setting. We assume that we are given a batch data set $\mathcal{D}$, in which each sample consists of a quadruple. The first two components are the state-action pair, corresponding to the state in which a given action was taken, and the last two components correspond to a noisy observation of the reward, and a successor state drawn from the appropriate transition function. Our theory allows for a very general dependence structure among the the state-action pairs in these samples; when the data set is ordered according to how the samples were collected (which need not be related to a trajectory), we allow the state-action pair at any given instant to depend on all past samples. This set-up allows from data collected from arbitrary policies, mixtures of policies, generative models or even in adversarial manner.

Given such a data set, our objective is to find the policy that performs best in the face of uncertainty. In particular, we need to account for the fact that the optimal policy $\pi^*$ for the underlying MDP may not be well covered by the dataset $\mathcal{D}$, in which case the associated uncertainty would be prohibitive. In order to achieve this goal, we design an actor-critic procedure that iteratively optimizes a lower bound on the value of the optimal policy. Suppose that we are interested in optimizing the value function at some given initial $s_1$. Our strategy works as follows: for any given policy $\pi$, we construct a family $\mathcal{M}(\pi)$ of "statistically plausible" MDPs, and use them to define a simple second-order cone program. By solving this convex program, we obtain value function estimate $\underline{V}_M^\pi(s_1) = \arg\min_{M \in \mathcal{M}(\pi)} V_M^\pi(s_1)$ that—for an appropriately constructed family $\mathcal{M}(\pi)$—is guaranteed to be a lower bound on the true value function of $\pi$ in the unknown MDP that generated the dataset. Given a procedure for producing such lower bounds, it is then natural to maximize these lower bounds over some family $\Pi$ of policies. This combination leads to the saddle-point problem

$$\max_{\pi \in \Pi} \min_{M \in \mathcal{M}(\pi)} V_M^\pi(s_1). \tag{1}$$

Note that actor-critic methods fit naturally in this framework: the critic provides a pessimistic evaluation of any given policy $\pi$, and the actor solves the outer maximization problem over policies.

---

[2]Exceptions to this include importance-sampling based approaches to selecting among a finite set of policies (e.g. Mandel et al. (2014); Thomas et al. (2015, 2019)); however, such approaches have focused on operating without a Markov assumption and inherently provide much looser guarantees than the ones we and others consider for the Markov setting.

This decoupling lends itself to a computationally tractable implementation, along with an analysis of the procedure. In particular, we show that the actor's sequence of estimated policies enjoys online learning-style guarantees with respect to a sequence of pessimistic MDPs implicitly identified by the critic.

The way in which we introduce pessimism is a second key component of the algorithmic framework. In particular, in line with our previous paper Zanette et al. (2020b), we do so without enlarging the prescribed classes of functions and policies. We do so by a direct perturbation of the value functions examined by the critic; there is no addition of pessimistic bonuses or absorbing states. Since the class of value functions is not altered, this method has two main advantages. First, there are no additional model assumptions compared to the standard—that is non-pessimistic—version of the actor-critic method. Second, the complexity of the underlying classes is not increased, thereby allowing us to construct tight confidence intervals and estimation error bounds that are minimax optimal up to logarithmic factors.

The remainder of this paper is organized as follows. We begin in Section 2 with background on MDPS, and then introduce the modeling assumptions that underlie the analysis of this paper. In Section 3, we introduce the algorithm studied in this paper, namely the Pessimistic Actor Critic for Learning without Exploration (for short, PACLE) algorithm. Section 4 provides statements of our main results and discussion of their consequences, including an upper bound on the PACLE algorithm in Theorem 1, and a minimax lower bound in Theorem 2. In Section A, we provide an outline of the proof of Theorem 1, with various technical details as well as the proof of Theorem 2 deferred to the appendices. We conclude with a discussion in Section 5.

## 1.2 Notation

For the reader's convenience, we summarize here some notation used throughout the paper. We let $\mathcal{B}_d(r) = \{x \in \mathbb{R}^d \mid \|x\|_2 \leq r\}$ denote the Euclidean ball of radius $r \in \mathbb{R}$ in dimension $d$; we simply write $\mathcal{B}$ when there is no possibility of confusion. For a vector $x \in \mathbb{R}^d$, we use $[x]_i$ to denote its $i^{th}$ component. We use the $\widetilde{O}$ notation to denote an upper bound that holds up to constants and log factors in the input parameters $(\frac{1}{\delta}, d, H)$. The notation $\lesssim$ means an upper bound that holds up to a constant, with an analogous definition for $\gtrsim$.

# 2 Background and problem formulation

We begin by providing some background, before introducing the assumptions that underlie our problem formulation.

## 2.1 Markov decision processes

In this paper, we focus on finite-horizon Markov decision processes, for which we provide a very brief introduction here. See the books Puterman (1994); Bertsekas and Tsitsiklis (1996); Bertsekas (1995); Sutton and Barto (2018) for more background and detail. A finite-horizon MDP is specified by a positive integer $H$, and events take place over a sequence of stages indexed by the time step $h \in [H] \overset{def}{=} \{1, \ldots, H\}$. The underlying dynamics involve a state space $\mathcal{S}$, and are controlled by actions that take values in some action set $\mathcal{A}$. In this paper, we allow the state space to be arbitrary (continous or discrete), whereas our analysis applies to discrete action spaces. For each time step $h \in [H]$, there is a reward function $r_h : \mathcal{S} \times \mathcal{A} \to \mathbb{R}$, and for every time step $h$ and state-action pair $(s, a)$, there is a transition function $\mathbb{P}_h(\cdot \mid s, a)$. When at horizon $h$, if the agent takes action $a$ in state $s$, it receives a random reward drawn from a distribution $R_h(s, a)$ with mean $r_h(s, a)$, and it then transitions randomly to a next state $s^+$ drawn from the transition function $\mathbb{P}_h(\cdot \mid s, a)$.

A policy $\pi_h$ at stage $h$ is a mapping from the state space $\mathcal{S}$ to the action space $\mathcal{A}$. Given a full policy $\pi = (\pi_1, \ldots, \pi_H)$, the state-action value function at time step $h$ is given by

$$Q_h^\pi(s, a) = r_h(s, a) + \mathbb{E}_{S_\ell \sim \pi \mid (s,a)} \sum_{\ell=h+1}^{H} r_\ell(S_\ell, \pi_\ell(S_\ell)), \qquad (2)$$

where the expectation is over the trajectories induced by $\pi$ upon starting from the pair $(s, a)$. When we omit the starting state-action pair $(s, a)$, the expectation is intended to start from a fixed state

denoted by $s_1$. The value function associated to $\pi$ is $V_h^\pi(s) = Q_h^\pi(s, \pi_h(s))$. For a given policy $\pi$, we define the Bellman evaluation operator

$$\mathcal{T}_h^\pi(Q_{h+1})(s, a) = r_h(s, a) + \mathbb{E}_{S' \sim \mathbb{P}_h(s,a)} \mathbb{E}_{A' \sim \pi} Q_{h+1}(S', A').$$

Under some regularity conditions Puterman (1994); Shreve and Bertsekas (1978), there always exists an optimal policy $\pi^\star$ whose value and action-value functions are defined as

$$V_h^\star(s) = V_h^{\pi^*}(s) = \sup_\pi V_h^\pi(s), \quad \text{and} \quad Q_h^\star(s, a) = Q_h^{\pi^*}(s, a) = \sup_\pi Q_h^\pi(s, a).$$

## 2.2 Assumptions on data generation

In this paper, we study a model in which we observe a dataset of the form $\mathcal{D} = \{(s_i, a_i, r_i, s_i^+)\}_{i=1}^n$, where $n$ is the total sample size. For each $i \in [n] = \{1, 2, \ldots, n\}$, the tuple $(s_i, a_i)$ corresponds to a state-action pair associated with some time step $h_i$. We let $\mathcal{F}_i$ be the $\sigma$-field generated by the samples $\{(s_j, a_j, r_j, s_j^+)\}_{j=1}^{i-1}$ that are in the "past" relative to index $i$. With this notation, we impose the following condition:

**Assumption 1** (Data generation). *For each $i \in [n]$, the pair $(s_i, a_i)$ is measurable with respect to $\mathcal{F}_i$. Conditionally on a given pair $(s_i, a_i)$, the random variable $r_i$ is drawn from a reward distribution $R_{h_i}(s_i, a_i)$ that is $1$-sub-Gaussian; and the next state $s_i^+$ is drawn from the distribution $\mathbb{P}_{h_i}(s_i, a_i)$.*

Note that the measurability condition allows the choice of $(s_i, a_i)$ to depend arbitrarily on any of the past data with indices $j < i$. The mild assumption allows for considerable freedom. For example, the state-action pairs may be chosen from (mixture) policies, or they can be generated by an adversarial procedure that changes the data acquisition strategy as feedback is received.

For each $h \in [H]$, we let $\mathcal{I}_h$ denote the subset of observation indices $i \in [n]$ such that $h_i = h$. These index sets define the sub-datasets $\mathcal{D}_h = \{(s_i, a_i, r_i, s_i^+), i \in \mathcal{I}_h\}$ associated with all samples that are based on state-action pairs at time step $h$. We define $n_h = |\mathcal{D}_h|$, so that our total sample size can be written as $n = \sum_{h=1}^H n_h$.

## 2.3 Policy and function classes

Next we define the policy space $\Pi$ and the action value function space $\mathcal{Q}$ over which we seek solutions. Let $\phi : \mathcal{S} \times \mathcal{A} \mapsto \mathbb{R}^d$ be a $d$-dimensional feature mapping. We assume throughout that these feature mappings are normalized such that $\|\phi(s, a)\|_2 \leq 1$ uniformly for all $(s, a)$-pairs. We consider action-value functions that are linear in $\phi$, and families of the form

$$\mathcal{Q}(\rho^w) \stackrel{def}{=} \{(s, a) \mapsto \langle \phi(s, a), w \rangle \mid \|w\|_2 \leq \rho^w\}, \tag{3a}$$

where $\rho^w \in (0, 1]$ is a user-defined radius. For policies, we consider the associated soft-max class

$$\Pi_{soft}(\rho^\theta) \stackrel{def}{=} \left\{ \frac{e^{\langle \phi(s,a), \theta \rangle}}{\sum_{a' \in \mathcal{A}} e^{\langle \phi(s,a'), \theta \rangle}} \mid \|\theta\|_2 \leq \rho^\theta \right\}, \tag{3b}$$

where $\rho^\theta > 0$ is a second radius.

In the context of our actor-critic algorithm, the weight radius $\rho^w$ remains fixed for all updates. On the other hand, the actor produces a sequence of soft-max radii $\{\rho_t^\theta\}_{t=1}^T$, indexed by the iterations $t$ of the actor. This sequence is produced via the update rule in Line 5 of Algorithm 1. The policy radius can be large $\rho^\theta \gg 1$ but we constrain $\rho^w \leq 1$ so that the critic's estimate $Q_w(s, a) = \langle \phi(s, a), w \rangle$ is bounded by one, i.e., $\sup_{(s,a,w)} |Q_w(s, a)| \leq 1$.

Recall that our MDP consists of sequence of $H$ distinct stages. Our algorithm and theory allows for the possibility of different feature extractors at each step $h \in [H]$, even with possibly different dimensions. Consequently, in implementing and analyzing the algorithm, there are actually $H$ (possibly different) functional spaces $\{\mathcal{Q}_h\}_{h=1}^H$, along with the associated soft-max policy classes $\{\Pi_h\}_{h=1}^H$. So as to simplify notation, we drop the dependence on the radii when referring to the functional spaces, and implicitly assume that the terminal value function is zero.

## 2.4 A range of function class assumptions

In this section, we discuss a range of assumptions that might be imposed on the class of action-value functions. This discussion serves as motivation for the particular assumption (Bellman restricted closedness—cf. Assumption 3) that underlies our analysis.

We begin with the least restrictive condition, which is a very natural starting point in our given set-up. If we seek to find the policy $\pi \in \Pi$ with the highest value function, it seems reasonable to require that the following representation condition (approximately) holds.

**Assumption 2** (Linear action-value functions $Q^\pi$). *The MDP admits a linear action-value function representation for all policies in $\Pi$, meaning that for each policy $\pi \in \Pi$ and time step $h \in [H]$, there exists a vector $w_h^\pi$ such that*

$$Q_h^\pi(s, a) = \langle \phi_h(s, a), w_h^\pi \rangle. \tag{4}$$

This assumption alone turns out to be inadequate to ensure that effective learning is possible; indeed, the recent papers Zanette (2020); Weisz et al. (2020) establish that even under this condition, there are instances that require exponentially many samples to do better than a random policy.

Given this fact, if one is interested in procedures with polynomial complexity (in both sample size and running time), stronger conditions need to be imposed. In general, the Bellman evaluation operator, even when applied to a linear action-value function, will return a nonlinear value function. The analysis of this paper is based on bounding the Bellman error in the sense of sup-norm deviation from linearity:

**Assumption 3** (Bellman Restricted Closedness). *The policy and value function spaces $(\Pi, \mathcal{Q})$ are closed up to $\nu \in \mathbb{R}^H$ error in the sup-norm if there is a non-negative sequence $\{\nu_h\}_{h=1}^H$ such that for each $h \in [H]$, we have*

$$\sup_{\substack{Q_{h+1} \in \mathcal{Q}_{h+1} \\ \pi_{h+1} \in \Pi_{h+1}}} \inf_{Q_h \in \mathcal{Q}_h} \|Q_h - \mathcal{T}_h^{\pi_{h+1}} Q_{h+1}\|_\infty \le \nu_h. \tag{5}$$

The restricted closedness assumption measures how well we can fit the action-value function resulting from the application of the Bellman evaluation operator to an action value function in $\mathcal{Q}$ and for a policy in $\Pi$. It enables the analysis of least-squares policy evaluation (e.g., Nedić and Bertsekas (2003)), which will be our starting point when constructing the critic.

Finally, for understanding connections to past work, it is relevant to compare to the *low-rank MDP* assumption that has been analyzed in recent work Jin et al. (2020a); Yang and Wang (2020), including in offline RL with pessimismistic guarantees Jin et al. (2020b), as well as in various online settings Agarwal et al. (2020a); Modi et al. (2021); Zanette et al. (2020a).

**Assumption 4** (Low-Rank MDP). *An MDP is low-rank if for all $h \in [H]$, there exists a reward parameter $w_h \in \mathbb{R}^d$ and a component-wise positive mapping $\psi_h : \mathcal{S} \to \mathbb{R}_+^d$ such that $\|\psi_h(s)\|_1 = 1$ for all $s \in \mathcal{S}$, and*

$$r_h(s, a) = \langle \phi_h(s, a), w_h \rangle, \qquad \mathbb{P}_h(s' \mid s, a) = \langle \phi_h(s, a), \psi_h(s') \rangle, \qquad \forall (s, a, h, s'). \tag{6}$$

The following proposition explicates the nested relationship between these three conditions, showing that the low-rank MDP condition is the most restrictive:

**Proposition 1** (Low Rank $\subset$ Restricted Closedness $\subset$ Linear $Q^\pi$). *For any fixed state-action space, horizon, and feature extractor:*

*(a) The class of low-rank MDPs is a strict subset of the class of MDPs that satisfy Bellman restricted closedness.*

*(b) The class of MDPs that satisfy Bellman restricted closedness is a strict subset of the linear $Q^\pi$ MDP class.*

See Appendix C for the proof of this claim.

Based on Proposition 1, we see that any analysis based on assuming Bellman restricted closedness also *a fortiori* applies to MDPs that satisfy the more stringent low-rank MDP condition.

## 3 The Pessimistic Actor-Critic

Given the set-up thus far, we are now ready to describe the actor-critic algorithm that we analyze in this paper. We refer to it as the *Pessimistic Actor Critic for Learning without Exploration*, or PACLE for short. We first describe the critic in Section 3.1, and then the actor in Section 3.2. We summarize the actor and critic algorithms, respectively, in pseudocode form in Algorithm 1 and Algorithm 2.

### 3.1 The Critic: Pessimistic Least Square Policy Evaluation

The purpose of the critic is to provide pessimistic value function estimates corresponding to the policy $\pi$ under consideration by the actor. Monte Carlo with importance sampling (IS) is not desirable in this setting, as the policy or distribution that generated the dataset might be unknown and estimation errors on the distribution can accumulate exponentially with the horizon in IS estimators (see e.g. Liu et al. (2018b)). Instead, we use a least-squares temporal difference method for policy evaluation, but suitably perturbed to return pessimistic estimates—i.e., lower bounds on the true value function of the given policy $\pi$. Our method is based on directly perturbing the regression parameters in the least-square estimate. In contrast to bonus-based approaches, this method has the important advantage of ensuring that the action-value function remains linear. The purpose of the perturbations is to compensate for possible statistical errors in estimating the regression parameter due to poor coverage of the given dataset.

Let us now give a precise description of the critic. Given a policy $\pi = (\pi_1, \ldots, \pi_H)$, the goal of the critic is to minimize the quantity

$$\mathbb{E}_{A' \sim \pi_1} \langle \phi(s_1, A'),\, w_1 \rangle = \sum_{a \in \mathcal{A}} \pi_1(a \mid s_1) \langle \phi_1(s_1, a),\, w_1 \rangle, \tag{7}$$

which is an estimate of the value function $V^\pi(s_1)$ for the policy $\pi$ at the initial state $s_1$. The parameter $w_1 \in \mathbb{R}^d$ is a vector to be adjusted, one that is determined by a backwards-running sequence of regression problems from $h = H$ down to $h = 1$.

We introduce the pessimistic perturbations directly to the solution of these regression problems. They involve a norm defined by the cumulative covariance matrix. Recall that $\mathcal{I}_h$ indexes the subset of observations associated with state-action pairs at time step $h$. For each $h \in [H]$ and $i \in \mathcal{I}_h$, let us write the associated sample as the quadruple $(s_{hi}, a_{hi}, r_{hi}, s_{h+1,i})$. Introducing the shorthand notation $\phi_{hi} = \phi_h(s_{hi}, a_{hi})$, we define the *cumulative covariance matrix*

$$\Sigma_h \stackrel{def}{=} \Big( \sum_{i \in \mathcal{I}_h} \phi_{hi} \phi_{hi}^\top \Big) + I_{d \times d}, \tag{8}$$

where $I_{d \times d}$ denotes the $d$-dimensional identity matrix. Notice that the cumulative covariance grows as the number of samples in $\mathcal{I}_h$ increases; we do not normalize it by the local sample size $n_h = |\mathcal{I}_h|$, so that $\Sigma_h$ effectively represents the amount of information contained in the sub-dataset $\mathcal{D}_h$ at time step $h$.

Since $\Sigma_h$ is strictly positive definite by construction, it defines a pair of norms

$$\|u\|_{\Sigma_h} \stackrel{def}{=} \sqrt{u^\top \Sigma_h u}, \quad \text{and} \quad \|u\|_{\Sigma_h^{-1}} \stackrel{def}{=} \sqrt{u^\top (\Sigma_h)^{-1} u}. \tag{9}$$

Consider the regression problem that is solved in moving backward from time step $h+1$ to $h$. Given the weight vector $w_{h+1}$ at time step $h + 1$, the regularized least-squares estimate of $w_h$ is given by

$$\widehat{w}_h \stackrel{def}{=} \Sigma_h^{-1} \sum_{k \in \mathcal{I}_h} \phi_{hk} \Big[ r_{hk} + \sum_{a \in \mathcal{A}} \pi_{h+1}(a \mid s_{h+1,k}) \langle \phi_{h+1}(s_{h+1,k}, a),\, w_{h+1} \rangle \Big].$$

We introduce pessimism by directly perturbing the weight vectors themselves—that is, we search for weight vector $w_h$ such that $w_h = \xi_h + \widehat{w}_h$, where the pessimism vector $\xi_h \in \mathbb{R}^d$ satisfies a bound of the form $\|\xi_h\|_{\Sigma_h} \leq \alpha_h$, for a user-defined parameter $\alpha_h$.

In detail, the critic takes as input the dataset $\mathcal{D}$, a policy $\pi$, a sequence of tolerance parameters $\alpha = (\alpha_1, \ldots, \alpha_H)$, weight radii $\rho^w = (\rho_1^w, \ldots, \rho_H^w)$ with each $\rho_h^w \in (0,1]$. The optimization variables consist of the regression vectors $w = (w_1, \ldots, w_H) \in (\mathbb{R}^d)^H$ and the pessimism vectors $\xi = (\xi_1, \ldots, \xi_H) \in (\mathbb{R}^d)^H$. The critic then solves the convex program

$$(\xi^\pi, \underline{w}^\pi) \overset{def}{=} \arg \min_{\substack{\xi \in (\mathbb{R}^d)^H \\ w \in (\mathbb{R}^d)^H}} \sum_{a \in \mathcal{A}} \pi_1(a \mid s_1) \langle \phi_1(s_1, a), w_1 \rangle \tag{10a}$$

with the terminal condition $w_{H+1} = 0$, and subject to the constraints

$$w_h = \xi_h + \Sigma_h^{-1} \sum_{k \in \mathcal{I}_h} \phi_{hk} \Big[ r_{hk} + \sum_{a \in \mathcal{A}} \pi_{h+1}(a \mid s_{h+1,k}) \langle \phi_{h+1}(s_{h+1,k}, a), w_{h+1} \rangle \Big], \qquad \text{and} \tag{10b}$$

$$\|\xi_h\|_{\Sigma_h}^2 \leq \alpha_h^2, \qquad \|w_h\|_2^2 \leq (\rho_h^w)^2 \tag{10c}$$

for all $h \in [H]$. Here the matrices $\Sigma_h$ were previously defined in equation (8).

The convex program (10) consists of a linear objective subject to quadratic constraints; it is a special case of a second order cone program, and can be efficiently solved with standard convex solvers.

---

**Algorithm 1** ACTOR (MIRROR DESCENT)
1: **Input**: Dataset $\mathcal{D}$, starting state $s_1$, learning rate $\eta$
2: Set $\theta_1 = (\vec{0}, \ldots, \vec{0})$
3: **for** $t = 1, 2, \ldots, T$ **do**
4: $\quad \underline{w}_t \leftarrow \text{CRITIC}(\mathcal{D}, \pi_{\theta_t}, s_1)$
5: $\quad \theta_{t+1} = \theta_t + \eta \underline{w}_t$
6: **end for**
7: **Return: Mixture policy** $\pi_{\theta_1}, \ldots, \pi_{\theta_T}$

**Algorithm 2** CRITIC (PLSPE)
1: **Input**: Dataset $\mathcal{D}$, target policy $\pi$, starting state $s_1$, critic radii $\{\rho_h^w\}_{h=1,\ldots,H}$, and parameters $\{\alpha_h\}_{h=1,\ldots,H}$
2: Solve the optimization program (10)
3: **Return:** Optimal weight vector $\underline{w}$

## 3.2 The Actor: Mirror Descent

We now turn to the behavior of the actor. It applies the mirror descent algorithm based on the Kullback Leibler (KL) divergence Bubeck (2014). This combination leads to the exponentiated gradient update rule in every timestep $h \in [H]$, so that the soft-max policy in moving from iteration $t$ to $t+1$ is updated as

$$\pi_{t+1,h}(a \mid s) \propto \pi_{t,h}(a \mid s) e^{\eta Q_h(s,a)} \qquad \text{for each } (s,a) \in \mathcal{S} \times \mathcal{A}. \tag{11}$$

Here $\eta > 0$ is a stepsize parameter, and our theory specifies a suitable choice.

If the $Q$-value above from the critic lives in $\mathcal{Q}$, then it is possible to show that $\pi_{t+1,h} \in \Pi_h$ and the update rule takes a much simpler and computationally more efficient form (cf. Line 5 of Algorithm 1), where $\underline{w}_t$ is the gradient of the value function on the pessimistic MDP implicitly identified by the critic. In this case, the spaces $(\mathcal{Q}, \Pi)$ are said to be *compatible* Sutton et al. (1999); Kakade (2001); Agarwal et al. (2020b); Raskutti and Mukherjee (2015) and the resulting algorithm is often called the *Natural Policy Gradient* (NPG) (see also Geist et al. (2019); Shani et al. (2020)). By construction, the critic maintains a linear action value function even after pessimistic perturbations. As a consequence, the actor policy space is the simple softmax policy class $\Pi$ and the easier update rule can be used. As we explain in the analysis, this has important statistical benefits.

After $T$ rounds of updates, the mirror descent algorithm that we use here readily achieves online regret rates (in the optimization setting with exact feedback) $\sim 1/T$ or $\sim 1/\sqrt{T}$ depending on the analysis Agarwal et al. (2020b) and the learning rate, although we mention that these rates could potentially be improved Khodadadian et al. (2021); Lan (2021); Bhandari and Russo (2020).

# 4 Main results

We now turn to the statement of a bound on the performance of the policy $\pi_{\text{ALG}}$ returned by PACLE. This upper bound involves three terms: an optimization error, an uncertainty term, and a model mis-specification term. The *optimization error* is given by $\mathcal{C}(T) \overset{def}{=} 4H\sqrt{\frac{\log|\mathcal{A}|}{T}}$; it captures the rate at which the error decreases as a function of the iterations of the actor. The *mis-specification error* $\mathcal{E}_{\text{msp}}(\nu) \overset{def}{=} \sum_{h=1}^{H} \nu_h$ is simply the sum of all the stage-wise mis-specification errors; notice that the mis-specification error does depend on the choice of the radii for the critic $\rho_1^w, \dots, \rho_H^w$ in a problem dependent way (cf. Assumption 3). Finally, for each $h$, define the vector $\bar{\phi}_h^\pi \overset{def}{=} \mathbb{E}_{(S_h, A_h) \sim \pi}[\phi_h(S_h, A_h)]$, where the expectation is over the state-action $(S_h, A_h)$ encountered at timestep $h$ upon following policy $\pi$. In terms of these vectors, the *uncertainty error* is given by

$$\mathcal{U}(\pi; \alpha) \overset{def}{=} 2\sum_{h=1}^{H} \alpha_h \|\bar{\phi}_h^\pi\|_{\Sigma_h^{-1}} = 2\sum_{h=1}^{H} \alpha_h \sqrt{(\bar{\phi}_h^\pi)^\top \Sigma_h^{-1} \bar{\phi}_h^\pi}, \tag{12}$$

where the cumulative covariance matrix $\Sigma_h$ was defined in equation (8).

The amount of information from the dataset $\mathcal{D}$ is fully encoded in the uncertainty function $\mathcal{U}$ through the sequence of cumulative covariance matrices $\{\Sigma_h\}_{h=1}^{H}$ and parameters $\{\alpha_h\}_{h=1}^{H}$. The more data are available, the more positive definite $\Sigma_h$ is and the smaller the uncertainty function $\mathcal{U}(\pi; \alpha)$ becomes for a fixed policy $\pi$. If the sampling distribution that generates the dataset is fixed, then we can write $\mathcal{U}(\pi; \alpha) \lesssim c/\sqrt{n}$ where $c$ does not depend on $n$ and can be interpreted as the coverage of the sampling distribution with respect to policy $\pi$.

## 4.1 A guarantee for PACLE

Our main result holds under Assumption 1 on the data collection process. It is based on radii $\{\rho_h^w\}_{h=1}^{H}$ for the action value function[3] that lie in the interval $(0, 1]$, and it provides a guarantee relative to the class $\Pi_{\text{all}}$ of all stochastic policies.

**Theorem 1** (An achievable guarantee). *Suppose that we are given a data set $\mathcal{D}$ collected in a way that respects Assumption 1. Then there are pessimism vectors bounded as $\alpha_h = \widetilde{O}(\sqrt{d \log(1/\delta)}) + \nu_h\sqrt{n_h}$ such that, after running $T \geq \log|\mathcal{A}|$ rounds of the actor with stepsize $\eta = \sqrt{\frac{\log|\mathcal{A}|}{T}}$, the PACLE procedure returns a policy $\pi_{\text{ALG}}$ for which*

$$V_1^\pi(s_1) - V_1^{\pi_{\text{ALG}}}(s_1) \leq \mathcal{U}(\pi; \alpha) + \underbrace{\sum_{h=1}^{H} \nu_h}_{\mathcal{E}_{msp}(\nu)} + \underbrace{4H\sqrt{\frac{\log|\mathcal{A}|}{T}}}_{\mathcal{C}(T)} \qquad \textit{uniformly over all } \pi \in \Pi_{all} \tag{13}$$

*with probability exceeding $1 - \delta$.*

The result provides a family of upper bounds on the sub-optimality of the learned policy $\pi_{\text{ALG}}$, indexed by the choice of comparator policy $\pi$, and embodies a tradeoff between the sub-optimality of the comparator $\pi$ and its uncertainty $\mathcal{U}(\pi; \alpha)$. Note that the optimization error $\mathcal{C}(T)$ can be reduced arbitrarily, while $\alpha$ (and thus $\mathcal{U}(\pi; \alpha)$) increase only logarithmically with $T$. As a special case, if we set $\pi = \pi^\star$ and assume that there is no mis-specification error, then we obtain that the learned policy satisfies a bound of the form

$$V_1^{\pi^\star}(s_1) - V_1^{\pi_{\text{ALG}}}(s_1) \leq \mathcal{U}(\pi^\star; \alpha) + \mathcal{C}(T) \tag{14}$$

with probability at least $1 - \delta$. Since $\mathcal{C}(T)$ is well-controlled, this guarantee is satisfied whenever the uncertainty term $\mathcal{U}(\pi^\star; \alpha)$ is small.

More generally, the guarantee (13) is significantly stronger than most prior work as PACLE competes not just with the optimal policy $\pi^\star$, but with all comparator policies simultaneously. Such comparator policies need not necessarily be in the prescribed policy class $\Pi$. To highlight the strength of this

---

[3]This represents a setting where both the reward and the value function can be as large as 1 in absolute value. One easily recovers the setting with value functions in $[0, H]$ using a rescaling argument.

generality, suppose that the uncertainty $\mathcal{U}(\pi^\star; \alpha)$ of the optimal $\pi^\star$ is *not* small—it could in fact be infinite. In this case, the bound (14) would not be useful.

However, suppose that there exists a near-optimal policy—meaning a policy $\pi^+$ such that $V_1^{\pi^+}(s_1) \geq V_1^\star(s_1) - \epsilon$ for some small $\epsilon$—that is well-covered by the dataset (i.e., for which $\mathcal{U}(\pi^+; \alpha) \approx 0$). In this case, Theorem 1 ensures with high probability $V_1^{\mathrm{ALG}}(s_1) \gtrsim V_1^\star(s_1) - \epsilon$. In contrast, traditional analyses that use only the optimal policy $\pi^\star$ as a comparator—as opposed to also allowing near-optimal policies—cannot return meaningful guarantees. We note also that the papers Yu et al. (2020); Liu et al. (2020); Kidambi et al. (2020) provide results of a similar flavor. These types of guarantees are also provided by some concurrent works Uehara and Sun (2021); Xie et al. (2021).

It should also be noted that Theorem 1 provides a family of results indexed by the choice of the critic's radii $\{\rho_h^w\}_{h=1}^H$. This choice is a modeling decision: increasing the radii increases both the approximation power of the function class $\mathcal{Q}_h$ used for regression, but also increases the complexity of the function class $\mathcal{Q}_{h+1}$ to represent (cf. Assumption 3); thus, the choice of the radii affects the approximation error $\mathcal{E}_{\mathrm{msp}}(\nu)$ in a problem dependent way.

## 4.2 A lower bound

Thus far, we have stated an upper bound on the quality of the returned policy for a given procedure. Central to this upper bound is the uncertainty function $\mathcal{U}(\pi; \alpha)$. In this section, we show that a term of this form is unavoidable for any procedure. In particular, working within the well-specified setting, we prove a lower bound in terms of the quantity $\mathcal{U}(\pi; \sqrt{d}) = \sqrt{d} \sum_{h=1}^H \|\bar\phi_h^\pi\|_{\Sigma_h^{-1}}$. Recalling that our choice of $\alpha$ scales with $\sqrt{d}$ (along with other logarithmic factors), this lower bound shows that our result is tight up to logarithmic factors.

We show that the lower bound actually holds in a setting that is easier for the learner, in the sense that (1) we restrict to low-rank MDPs, where there is no mis-specification error; and (2) the mechanism that generates the dataset is non-adaptive, and so certainly satisfies Assumption 1.

**Theorem 2** (Information-theoretic lower bound). *For a given horizon $H$ and dimension $d$, consider a sample size $n \geq 2d^3H^3$. There is a class $\mathcal{M}$ of low-rank MDPs and a data generating procedure satisfying Assumption 1 such that for any policy $\widehat\pi_{\mathrm{ALG}}$, we have*

$$\sup_{M \in \mathcal{M}} \mathbb{E}_M \left[ V_{1M}^\pi(s_1) - V_{1M}^{\widehat\pi_{\mathrm{ALG}}}(s_1) \right] \geq c\, \mathcal{U}(\pi; \sqrt{d}) \qquad \textit{uniformly over all } \pi \in \Pi_{\mathit{all}}, \qquad (15)$$

*where $c > 0$ is a universal constant.*

When $H = 1$ the above result gives a sample complexity lower bound for learning a near optimal policy from batch data in a linear bandit instance.

## 4.3 Comparison to related work

Theorem 1 automatically implies the typical bound $\mathbb{P}[V_1^{\pi_{\mathrm{ALG}}}(s_1) \geq V_1^\star(s_1) - \mathcal{U}(\pi^\star; \alpha)] \geq 1 - \delta$ when the comparator policy is the optimal policy $\pi^\star$, e.g., Jin et al. (2020b); Rashidinejad et al. (2021); Kidambi et al. (2020); Kumar et al. (2019); Buckman et al. (2020). The guarantee can be written as $V_1^{\pi_{\mathrm{ALG}}}(s_1) \gtrsim V_1^\star(s_1) - C/\sqrt{n}$ where $n$ is the number of samples and $C$ is the (scaled) condition number of $\Sigma_h^{-1}$. One could interpret $C$ as a concentrability coefficient that expresses the coverage of dataset—through $\Sigma_h$—with respect to the average direction in feature space $\mathbb{E}_{(S_h, A_h) \sim \pi_h^\star}[\phi(S_h, A_h)]$ of the optimal policy $\pi^\star$. As in the paper Jin et al. (2020b), such a factor can be small even when traditional concentrability coefficients are large because they depend on state-action visit ratios (see the literature in Appendix B, e.g., Chen and Jiang (2019)).

With reference to the results in the paper Jin et al. (2020b), our work provides improvements in two distinct ways. First, their upper and lower bounds exhibit a gap of the order $dH$, which our analysis closes. Second, our analysis holds under the more permissive Assumption 3 (*Bellman Restricted Closedness*) which includes low-rank MDPs. Of this improvement, a factor of $\sqrt{d}$ is due to the algorithm that we use, and the remainder is due to a more refined construction to certify optimality in Theorem 2. To be clear, our upper and lower bounds differ from theirs by a factor of $H$ due to a different normalization in the value function). We also note that the result of Liu et al. Liu et al.

(2020) can be specialized to the low-rank MDP setting; however, even in this simpler setting, the results would be sub-optimal and also require additional density estimates.

Deriving a computationally tractable model-free algorithm without low-rank dynamics but subject to value function perturbations (e.g., optimistic or pessimistic perturbations) is an open problem even in the more heavily studied online exploration setting: there the current state-of-the art Zanette et al. (2020b); Jin et al. (2021); Du et al. (2021); Jiang et al. (2017) only present computationally *intractable* algorithms with the exception of Zanette et al. (2020c) for a PAC setting with low inherent Bellman error which however requires an additional "explorability" condition. Due to space constraints, the proof outline is deferred to Appendix A.

## 5 Discussion

In this paper, we have developed and analyzed an actor-critic method procedure, designed for finding near-optimal policies in the offline setting. The PACLE procedure introduces pessimism into the critic's evaluation of a given policy's value function, thereby ensuring that, under suitable parameter choices and assumptions, it maintains (with high probability) a lower bound on the true value function. The actor then performs a form of mirror ascent so as to maximize the value of these lower bounds.

An important feature of our method is that it introduces pessimism via direct perturbations of the parameter vectors in a linear function approximation scheme. In this way, we avoid having to impose additional model assumptions; moreover, the pessimism does *not* substantially increase the complexity of our under value/policy classes, which allows us to provide minimax-optimal guarantees. We note that similar approaches have appeared before in the exploration setting; for example, see the recent papers Zanette et al. (2020b); Jin et al. (2021); Du et al. (2021). These methods enjoy similar advantages in terms of theoretical guarantees, but at the expense of computational tractability. In contrast, the method of this paper entails solving a low-dimensional second-order cone program, a simple class of convex programs for which there exist many polynomial-time algorithms. We enjoy this advantage due to some key differences between the offline and online settings of RL. In the offline setting, it is possible to keep the actor's update cleanly separated from the evaluation step of the critic, as we have done here; this separation underlies the computational tractability.

Our work leaves open a number of interesting questions for future work. First, it would be interesting to provide some numerical studies of the PACLE's performance, so as to understand its practical behavior relative to the theoretical guarantees provided here. Also, our analysis here has focused purely on approximation using linear basis expansions; extension to more general function classes is an important next step. Finally, it will be interesting to see to what extent these ideas can be translated to the more challenging setting of exploration.

### Acknowledgements

This work was partially supported by NSF-DMS grant 2015454, NSF-IIS grant 1909365, and NSF-FODSI grant 2023505 to MJW, a Stanford Artificial Intelligence Laboratory Toyota gift to EB, and a Office of Naval Research grant DOD-ONR-N00014-18-1-2640 to MJW.

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
