1. The main components of the proof are guarantees for the pessimistic estimates produced by the critic, and online learning guarantees for the updates taken by the actor. These two guarantees are coupled together via the notion of an induced MDP.

The proof outline given here follows a bottom-up approach: (a) starting with the critic in Section A.1, we first introduce the notion of induced MDP that links the critic's output to the actor's input (see Section A.1.1), and then discuss how suitable choices of the pessimism parameters $\alpha$ allow us to guarantee that the critic underestimates the true value function (see Sections A.1.2 and A.1.3); (b) next in Section A.2, we provide online-style learning guarantees for the actor, again using the notion of induced MDP to link these guarantees back to the critic; and (c) in Section A.3, we put together the pieces to prove the theorem itself.

## A.1 Critic's Analysis

Given a policy $\pi$ and pessimism parameters $\alpha$ for which the convex program (10) is feasible, the critic returns the pair $(\underline{\xi}^\pi, \underline{w}^\pi) = \{(\underline{\xi}_h^\pi, \underline{w}_h^\pi)\}_{h=1}^H$. These weight vectors induce the estimated value functions

$$\underline{Q}_h^\pi(s,a) \stackrel{def}{=} \langle \phi(s,a), \underline{w}_h^\pi \rangle, \qquad \text{and} \quad \underline{V}_h^\pi(s) \stackrel{def}{=} \mathbb{E}_{A' \sim \pi_h(\cdot|s)} \underline{Q}_h^\pi(s, A'). \tag{16}$$

Our goal in analyzing the critic is to relate these critic-estimated value functions to the true value functions $\{Q_h^\pi\}_{h=1}^H$.

### A.1.1 Induced MDP

Essential to our analysis is an object that provides the essential link between the critic's output and the actor's input. In particular, it is helpful to understand the critic in the following way: when given a policy $\pi$ as input, the critic computes the estimates $\{\underline{Q}_h^\pi\}_{h=1}^H$, and uses them form a new MDP $\hat{M}(\pi)$, which we refer to as the *induced MDP*. This new MDP shares the same state/action space and transition dynamics with the original MDP $M$, differing only in the perturbation of the reward function. In particular, for each $h \in [H]$, we define the *perturbed reward function*

$$\widehat{r}_h^\pi(s,a) \stackrel{def}{=} r_h(s,a) + \underline{Q}_h^\pi(s,a) - \mathcal{T}_h^\pi(\underline{Q}_{h+1}^\pi)(s,a). \tag{17}$$

The induced MDP $\hat{M}(\pi)$ is simply the original MDP that uses this perturbed reward function.

One important property of the induced MDP—which motivates the definition (17)—is that the estimates (16) returned by the critic correspond to the *exact value functions* of policy $\pi$ in the induced MDP. We summarize in the following:

**Lemma 1** (Critic exactness in induced MDP). *Given a policy $\pi$ as input, the critic returns a sequence $\{\underline{V}_h^\pi\}_{h=1}^H$ such that*

$$\underline{Q}_h^\pi = Q_{h,\hat{M}(\pi)}^\pi, \quad \text{and} \tag{18a}$$

$$\underline{V}_h^\pi = V_{h,\hat{M}(\pi)}^\pi \qquad \text{for all } h \in [H], \tag{18b}$$

*where $V_{h,\hat{M}(\pi)}^\pi$ is the exact value function of policy $\pi$ in the induced MDP $\hat{M}(\pi)$.*

See Section D.1 for the proof of this claim.

Moreover, since the induced MDP differs from the original MDP only in terms of the reward perturbation (17), we have the following convenient property: for any policy $\widetilde{\pi}$—which need not be of the soft-max form—the definition of value functions ensures that

$$V_{1,\hat{M}(\pi)}^{\widetilde{\pi}}(s_1) - V_1^{\widetilde{\pi}}(s_1) = \sum_{h=1}^H \mathbb{E}_{(S_h, A_h) \sim \widetilde{\pi}} \left[ \widehat{r}_h^\pi(S_h, A_h) - r_h(S_h, A_h) \right], \tag{19}$$

where $V_{1,\hat{M}(\pi)}^{\widetilde{\pi}}$ is the value function of $\widetilde{\pi}$ in the induced MDP. This simple relation allows us to use the induced MDP to relate arbitrary policies to their exact value functions.

### A.1.2 Critic's guarantee under a "good" event

We now show that there is a "good event"—call it $\mathcal{G}(\alpha)$—under which the critic's value function estimates have some additional desirable properties. Once this event is defined, the core of our proof involves determining the smallest choice of pessimism parameters under which it holds with probability at least $1 - \delta$.

We begin with some notation required to define the good event. Let $\mathcal{F}$ denote the space of all real-valued functions on $\mathcal{S} \times \mathcal{A}$. The *regression operator* is a mapping from $\mathcal{F}$ to $\mathbb{R}^d$, given by

$$\mathcal{R}_h^\pi(F) \overset{def}{=} \Sigma_h^{-1} \sum_{k=1}^T \phi_{hk} \big\{ r_{hk} + \mathbb{E}_{A' \sim \pi(\cdot|s_{hk})} F(s_{h+1,k}, A') \big\}, \tag{20a}$$

where $F \in \mathcal{F}$. To appreciate the relevance of the regression operator, note that by definition of the critic, we have the equivalence

$$\underline{w}_h^\pi = \underline{\xi}_h^\pi + \mathcal{R}_h^\pi(\underline{Q}_{h+1}^\pi). \tag{20b}$$

We also define the *sup-norm projection operator* (for the definition of $\mathcal{B}$ please see Section 1.2)

$$\mathcal{P}_h^\pi(F) \overset{def}{=} \arg \min_{w_h \in \mathcal{B}(\rho_h^w)} \sup_{(s,a)} \Big| \langle \phi(s,a), w_h \rangle - (\mathcal{T}_h^\pi F)(s,a) \Big|. \tag{20c}$$

Note that $\mathcal{P}_h^\pi$ is a mapping from $\mathcal{F}$ to $\mathbb{R}^d$; it returns the weight vector of the best-fitting linear function to the Bellman update $\mathcal{T}_h^\pi(F)$.

Our good event is defined in terms of the *parameter error operators* $\mathcal{E}_h^\pi : \mathcal{F} \to \mathbb{R}^d$ given by

$$\mathcal{E}_h^\pi(F) \overset{def}{=} \mathcal{R}_h^\pi(F) - \mathcal{P}_h^\pi(F). \tag{21}$$

For a given sequence $\alpha = (\alpha_1, \ldots, \alpha_H)$ of pessimism parameters, we define the *good event*

$$\mathcal{G}(\alpha) \overset{def}{=} \Big\{ \sup_{\substack{Q_{h+1} \in \mathcal{Q}_{h+1} \\ \pi_{h+1} \in \Pi_{h+1}}} \| \mathcal{E}_h^{\pi_{h+1}}(Q_{h+1}) \|_{\Sigma_h} \leq \alpha_h \quad \text{for all } h \in [H] \Big\}. \tag{22}$$

**Some intuition:** Why is this event relevant for guaranteeing good performance of the critic? In order to gain intuition, let us consider the special case in which there is no approximation error, so that the exact state-action value functions are actually linear. Letting $w_h^\pi$ denote the parameter associated with the linear action-value function at step $h$, when the good event holds, our choice of $\alpha$ allows us to set

$$\underline{\xi}_h^\pi = -\mathcal{E}_h^{\pi_{h+1}}(Q_{h+1}^\pi) = w_h^\pi - \mathcal{R}_h^\pi(Q_{h+1}^\pi) \qquad \text{for each } h \in [H],$$

in the constraints (10b). In this way, at each step $h$ the vector $\underline{\xi}_h^\pi$ can perfectly compensate the noise error $\mathcal{E}_h^{\pi_{h+1}}(Q_{h+1}^\pi)$ ensuring that the action-value function $Q_h^\pi$ (compactly encoded in the parameter $w_h^\pi$) can be perfectly represented. In other words, our choice guarantees that the feasible set for (10) contains the 'true' solution $w_h^\pi$. Since the convex program involves minimizing over value functions, this feasibility underlies showing the critic returns an underestimate of the true value function for $\pi$ along with some approximation error in the general setting; see equation (23a) below for a precise statement. We highlight that such underestimates is only guaranteed at the initial state $s_1$ and timestep $h = 1$ as encoded in the objective of the program in equation (10).

On the other hand, for other policies $\widetilde{\pi}$, we can use the relation (19) to control the difference between the value function $V_{1,\hat{M}(\pi)}^{\widetilde{\pi}}$ in the induced MDP, and the exact value function $V_1^{\widetilde{\pi}}$; see equation (23b) for a precise statement of our conclusion. We summarize all of our findings thus far in the following:

**Proposition 2.** *Conditionally on the event $\mathcal{G}(\alpha)$, when given as input any policy $\pi$ in the soft-max class $\Pi_{soft}(R)$, the critic returns an induced MDP $\hat{M}(\pi)$ such that:*

   *(a) For the given policy $\pi$, we have*

$$V_{1,\hat{M}(\pi)}^\pi(s_1) \leq V_1^\pi(s_1) + \sum_{h=1}^H \nu_h. \tag{23a}$$

*(b) For any policy $\widetilde{\pi}$, not necessarily in the soft-max class $\Pi$, we have*

$$\left| V_{1,\hat{M}(\pi)}^{\widetilde{\pi}}(s_1) - V_1^{\widetilde{\pi}}(s_1) \right| \leq 2 \sum_{h=1}^{H} \alpha_h \, \|\bar{\phi}_h^{\widetilde{\pi}}\|_{\Sigma_h^{-1}} + \sum_{h=1}^{H} \nu_h, \tag{23b}$$

*where $\bar{\phi}_h^{\widetilde{\pi}} \overset{def}{=} \mathbb{E}_{(S_h,A_h)\sim\widetilde{\pi}}[\phi_h(S_h, A_h)]$.*

See Appendix D.2 for the proof.

### A.1.3 Choice of pessimism parameters

Based on Proposition 2, our problem is now reduced to determining a choice of $\alpha$ for which the good event (22) holds with probability at least $1 - \delta$. The bulk of our effort in analyzing the critic is devoted to the technical details of this step; we provide only a high-level summary here.

The event (22) needs to hold uniformly over the value function and policy classes used by the algorithm. Our analysis involves deriving an upper bound $R$ on the $\ell_2$-radius of the actor parameter over all $T$ iterations of the algorithm, as follows:

$$\rho^\theta = \|\theta_T\|_2 = \|\sum_{t=1}^{T} \eta w_t\|_2 \leq \sum_{t=1}^{T} \eta \|w_t\|_2 \leq T\eta \overset{def}{=} R.$$

For such choice of $R$ and failure probability $\delta \in (0,1)$, suppose that we set

$$\alpha_h(\delta) \overset{def}{=} 1 + \sqrt{n_h}\nu_h + c\left\{1 + d\log\left(1 + \tfrac{T}{d}\right) + d\log\left(1 + 8\sqrt{T}\right) + d\log\left(1 + 16R\sqrt{T}\right) + \log\tfrac{H}{\delta}\right\}^{1/2} \tag{24}$$

for a suitably large universal constant $c$. Central to our analysis is the following lemma:

**Lemma 2.** *For any $\delta \in (0,1)$, given the choice of pessimism vector $\alpha(\delta)$ in equation (24), we have*

$$\mathbb{P}\big[\mathcal{G}(\alpha(\delta))\big] \geq 1 - \delta. \tag{25}$$

See Section D.3 for the proof of this claim.

In our proof of Lemma 2, we benefit from the fact that our procedure injects its pessimism by direct perturbations of the parameter vectors. Indeed, one key step in the proof is bounding certain metric entropies defined by classes $\mathcal{Q}$ of linear action-value functions, and policy classes $\Pi_{soft}(R)$ used in the actor's iterations.

First, for any fixed policy $\pi$, since the agent's action value function $\underline{Q}^\pi$ is enforced to be linear $\underline{Q}^\pi \in \mathcal{Q}$ even after perturbations, the relevant action-value class $\mathcal{Q}$ is also linear. Thus, we need only control metric entropy (and perform union bounds over the resulting covering) for a linear function class; in this way, we avoid a potentially more costly union bound over the much larger function class obtained by adding complex bonuses to linear functions, as in past work Jin et al. (2020b). In this way, we achieve a guarantee that is sharper by a factor of $\sqrt{d}$.

Second, the union bound needs to be extended to all policies that the actor can use to invoke the critic. Recall that the critic returns a linear action-value function $\underline{Q}$, which is compatible Kakade (2001); Agarwal et al. (2020b) with the soft-max policy class $\Pi_{soft}$. Consequently, the actor's updates take the simple form (5) of Algorithm 1. If the action-value function $\underline{Q}$ were perturbed by bonuses, then linearity of the critic's value function would be lost.

### A.2 Actor's Analysis

In this section, we analyze the mirror descent algorithm—that is, the actor in Algorithm 1. Our analysis exploits the methods in the paper Agarwal et al. (2020b), with some small changes to accommodate our framework; in particular, while our analysis assumes no error in the critic's evaluation, it does involve a sequence of time-varying MDPs.

Given a sequence of MDPs $\{M_t\}_{t=1}^{T}$, let $V_t^\pi$ be the value function associated with policy $\pi$ on MDP $M_t$. Given the initialization $\theta_1 = 0$, let $\{\theta_t\}_{t=1}^{T}$ be parameter sequence generated by the

actor, and let $\pi_t = \pi_{\theta_t}$ be the policy associated with parameter $\theta_t$. For each $t$, there is a sequence $w_t = \{w_{ht}\}_{h=1}^H$ such that $\|w_{ht}\|_2 \le \rho_h^w$ for all $h \in [H]$, and

$$Q_{h,M_t}^{\pi_t}(s,a) \stackrel{def}{=} \langle \phi_h(s,a), w_{ht} \rangle, \qquad \text{for all } (s,a) \text{ and } h \in [H]. \tag{26a}$$

In particular, the value of $w_{ht}$ is the value $\underline{w}_{ht}$ identified by the critic (see equation (16)) corresponding to policy $\pi_t$, so that $Q_{M_t}^{\pi_t} = \underline{Q}^{\pi_t}$. Define the value function $V_{h,M_t}^{\pi_t}(s) = \mathbb{E}_{A' \sim \pi_t}\left[Q_{h,M_t}^{\pi_t}(s,A')\right]$ along with the advantage function

$$G_{h,M_t}^{\pi_t}(s,a) \stackrel{def}{=} Q_{h,M_t}^{\pi_t}(s,a) - V_{h,M_t}^{\pi_t}(s). \tag{26b}$$

**Proposition 3** (Actor's Analysis). *Suppose that the actor takes $T \ge \log|\mathcal{A}|$ steps using a stepsize $\eta \in (0,1)$, and the advantage function at each iteration $t$ is uniformly bounded as $|G_{h,M_t}^{\pi_t}(s,a)| \le 2$ for all $(s,a)$. Then for any fixed policy $\pi$, we have*

$$\frac{1}{T}\sum_{t=1}^T \left\{ V_{1,M_t}^{\pi}(s_1) - V_{1,M_t}^{\pi_t}(s_1) \right\} \le H\left[\frac{\log|\mathcal{A}|}{\eta T} + \eta\right]. \tag{27a}$$

*In particular, setting $\eta = \sqrt{\frac{\log|\mathcal{A}|}{T}}$ yields the bound*

$$\frac{1}{T}\sum_{t=1}^T \left\{ V_{1,M_t}^{\pi}(s_1) - V_{1,M_t}^{\pi_t}(s_1) \right\} \le \underbrace{2H\sqrt{\frac{\log|\mathcal{A}|}{T}}}_{=\mathcal{C}(T)}. \tag{27b}$$

To be clear, the fixed comparator policy $\pi$ in the above bounds need not be in $\Pi$. This fact is important, as it allows us to derive bounds relative to an arbitrary comparator.

## A.3 Combining the pieces

We are now ready to combine the pieces so as to prove Theorem 1. For each iteration $t \in [T]$, let $\pi_t \stackrel{def}{=} \pi_{\theta_t}$ be the policy chosen by the actor, and let $M_t = M_{\pi_t}$ be the corresponding induced MDP.

Recall that Lemma 2, stated in Section D.2, guarantees that the "good" event $\mathcal{G}$ from equation (22) occurs with probability at least $1-\delta$. Conditioned on the occurrence of $\mathcal{G}$, the bounds (23a) and (23b) ensure that for any comparator $\widetilde{\pi}$, we have

$$V_1^{\widetilde{\pi}}(s_1) - V_1^{\pi_t}(s_1) \le V_{1,M_t}^{\widetilde{\pi}}(s_1) - V_{1,M_t}^{\pi_t}(s_1) + 2\sum_{h=1}^H \left[\nu_h + \alpha_h\|\mathbb{E}_{(S_h,A_h)\sim\widetilde{\pi}}\phi(S_h,A_h)\|_{\Sigma_h^{-1}}\right]$$

$$= V_{1,M_t}^{\widetilde{\pi}}(s_1) - V_{1,M_t}^{\pi_t}(s_1) + \mathcal{E}_{\mathrm{msp}}(\nu) + \mathcal{U}(\widetilde{\pi};\alpha).$$

We now average over the iterations $t \in [T]$. The equality (18a) from Lemma 1 ensures for each iteration $t$, the actor receives as an input a vector $\underline{w}_t$ such that

$$Q_{h,M_t}^{\pi_t}(s,a) \stackrel{Lem.1}{=} \underline{Q}_h^{\pi_t}(s,a) = \langle \phi_h(s,a), \underline{w}_{hk} \rangle. \tag{28}$$

Consequently, the action-value function $\underline{Q}^{\pi_t}$ that provided as input to the actor via $\underline{w}_t$ is the action-value function of $\pi_t$ on the associated induced MDP $M_t$, i.e., $Q_{M_t}^{\pi_t}$. Applying the bound (27b) from Proposition 3 yields $\frac{1}{T}\sum_{t=1}^T \left[V_{1,M_t}^{\widetilde{\pi}}(s_1) - V_{1,M_t}^{\pi_t}(s_1)\right] \le \mathcal{C}(T)$. Combining with the prior display yields

$$V_1^{\widetilde{\pi}}(s_1) - \frac{1}{T}\sum_{t=1}^T V_1^{\pi_t}(s_1) \le \mathcal{C}(T) + \mathcal{E}_{\mathrm{msp}}(\nu) + \mathcal{U}(\widetilde{\pi};\alpha). \tag{29}$$

Notice that the policy returned by the agent $\pi_{\mathrm{ALG}}$ is the mixture policy of the policies $\pi_1,\ldots,\pi_T$ and its value function is $V^{\pi_{\mathrm{ALG}}} = \frac{1}{T}\sum_{t=1}^T V^{\pi_t}$.

Note that under the good event $\mathcal{G}$, the bound (29) holds for any comparator policy $\widetilde{\pi}$, which was the claim of the theorem.

## B   Additional Literature

For empirical studies on offline RL, see the papers Laroche et al. (2019); Jaques et al. (2019); Wu et al. (2019); Agarwal et al. (2020c); Wang et al. (2020); Siegel et al. (2020); Nair et al. (2020) in addition to those presented in the main text. Several works have investigated offline policy learning, where concentrability coefficients are introduced to account for the non-uniform error propagation Munos (2003, 2005); Antos et al. (2007, 2008); Farahmand et al. (2010, 2016); Chen and Jiang (2019); Xie and Jiang (2020a,b); Duan et al. (2021). For additional literature, see also the papers Zhang et al. (2020a); Liao et al. (2020); Fan et al. (2020); Fu et al. (2020); Wang et al. (2019). Concentrability coefficients or density ratios also appears in the off-policy evaluation problem, which is distinct from the policy learning problem that we consider here Zhang et al. (2020b); Thomas and Brunskill (2016); Farajtabar et al. (2018); Liu et al. (2018a); Xie et al. (2019); Yang et al. (2020); Nachum et al. (2019b); Yin et al. (2020); Yin and Wang (2020); Duan and Wang (2020); Uehara et al. (2020); Jiang and Huang (2020); Kallus and Uehara (2019); Tang et al. (2019); Nachum and Dai (2020); Nachum et al. (2019a); Jiang and Li (2016); Uehara et al. (2020); Voloshin et al. (2021); Jiang and Huang (2020); Hao et al. (2021).

## C   Proof of Proposition 1

First, let us define an MDP class indexed by $N$; we will use this MDP class to show that each inclusion is strict. At a high level, this MDP class has a starting state 0 where the agent can choose to go left (action $-1$) or right (action $+1$); after that, it will keep going left or right until the leftmost or rightmost terminal state is reached. The reward is non-zero only at the terminal states.

For a fixed $N$, let the horizon be $H = N + 1$ and consider the following chain MDP, where the state space is

$$\mathcal{S} = \{N, -(N-1), \ldots, -1, 0, +1, \ldots, N-1, N\}.$$

The starting state is 0, and there the agent can choose among two actions ($-1$ and $+1$). In states $s \neq 0$ only one action is available. Formally, we define

$$\mathcal{A}_s = \begin{cases} \{-1\} & \text{if } s < 0 \\ \{-1, +1\} & \text{if } s = 0 \\ \{+1\} & \text{if } s > 0. \end{cases} \tag{30}$$

The reward is everywhere zero except in the terminal states $-N$ and $+N$, for which it takes the values $-1$ and $+1$, respectively, for the only action available there. The transition function is deterministic, and the successor state is always $s' = s + a$ (e.g., action $+1$ in state $+2$ leads to state $+3$). In other words, if the agent is a state $s$ with positive value, it will move to $s+1$, and if $s$ has negative value it will move to $s - 1$.

### C.1   Proof of part (a): Low Rank $\subset$ Restricted Closed

We first prove that a low-rank MDP must satisfy the restricted closedness assumption. Assume the MDP is low rank. Then for any $Q_{h+1} \in \mathcal{Q}_{h+1}$ and $\pi \in \Pi$, we have

$$\begin{aligned} \mathcal{T}_h^\pi Q_{h+1} &= \left\langle \phi_h(s,a), w_h^R \right\rangle + \left\langle \phi_h(s,a), \int_{s'} \mathbb{E}_{a' \sim \pi} Q_{h+1}(s', a') d\psi(s') \right\rangle \\ &= \left\langle \phi_h(s,a), w_h^R + \int_{s'} \mathbb{E}_{a' \sim \pi} Q_{h+1}(s', a') d\psi(s') \right\rangle \\ &= \langle \phi_h(s,a), w \rangle \end{aligned}$$

for some $w \in \mathbb{R}^d$. Thus, we have $(\mathcal{T}_h^\pi Q_{h+1}) \in \mathcal{Q}_h$ for all $Q_{h+1} \in \mathcal{Q}_{h+1}$ and $\pi \in \Pi$—i.e., if the MDP is low rank then it satisfies the restricted closedness condition.

In order to establish the strict inclusion, consider the MDP described at the beginning of the proof with the following feature extractor:

$$\phi(s,a) = \begin{cases} +1 & \text{if } a = +1 \\ -1 & \text{if } a = -1. \end{cases} \tag{31}$$

The MDP with this feature map is not low rank. For example, we must have

$$1 = \mathbb{P}(N \mid N - 1, +1) = \phi(N - 1, +1)^\top \psi(N) = \psi(N)$$

which implies $\psi(-N) = 0$ for $\psi$ to be a measure. However, this means we won't be able to represent all transitions correctly, as we would need to have

$$1 = \mathbb{P}(-N \mid -(N - 1), -1) = \phi(-(N - 1), -1)^\top \psi(-N) = -\psi(-N) = 0.$$

This means the MDP is not low rank. However, we show that it still satisfies the restricted closedness assumption. Notice that it is enough to verify the condition in the reachable space, which is $|s| + 1 = h$ at timestep $h$. If the reward is zero it suffices to verify that for all choices of $\theta_{h+1}$ we can find $\theta_h$ such that

$$\langle \phi(h - 1, +1), \theta_h \rangle = \langle \phi(h, +1), \theta_{h+1} \rangle \tag{32}$$

$$\langle \phi(-(h - 1), -1), \theta_h \rangle = \langle \phi(-h, -1), \theta_{h+1} \rangle . \tag{33}$$

Notice that in all cases there is only one policy available at the successor states; for any choice of $\theta_{h+1}$, just set $\theta_h = \theta_{h+1}$. It is easy to verify that at the last step $h = H = N + 1$ the reward function is either $+1$ or $-1$, depending on the state, and can be represented by $\theta_h = +1$:

$$\langle \phi(H - 1, +1), \theta_H \rangle = +1 \tag{34}$$

$$\langle \phi(-(H - 1), -1), \theta_H \rangle = -1. \tag{35}$$

### C.2 Proof of part (b): Restricted Closedness $\subset$ Linear $Q^\pi$

We first show that every MDP that satisfies restricted closedness satisfies the linear $Q^\pi$ assumption. For any time step $h \in H$, and for a given policy $\pi \in \Pi$, if restricted closedness holds, choose $Q_{h+1} = Q_{h+1}^\pi$ in the definition of restricted closedness and use the Bellman equations to obtain

$$Q_h^\pi \overset{def}{=} \mathcal{T}_h^\pi Q_{h+1}^\pi \in \mathcal{Q}_h.$$

Thus, the linear $Q^\pi$ assumption is automatically satisfied.

In order to show the strict inclusion, consider again the MDP described at the beginning of the proof, but with a different feature map. The map reads

$$\phi(s, a) = \begin{cases} [+1, 0] & \text{if } a = +1, s \neq 0 \\ [0, +1] & \text{if } a = -1, s \neq 0, \end{cases}$$

and at the start state

$$\phi(0, a) = \begin{cases} +1 & \text{if } a = +1 \\ -1 & \text{if } a = -1. \end{cases}$$

Notice that we only need to verify that restricted closedness does not hold at some timestep. When $\theta_2 = [+1, +1]$, there is no $\theta_1$ such that

$$+\theta_1 = \langle \phi(0, +1), \theta_1 \rangle = \langle \phi(1, 1), \theta_2 \rangle = 1$$

$$-\theta_1 = \langle \phi(0, -1), \theta_1 \rangle = \langle \phi(-1, -1), \theta_2 \rangle = 1.$$

The MDP however satisfies the linear $Q^\pi$ assumption with $\theta_1 = 1$ and $\theta_h = [+1, -1]$ for $h \geq 2$.

## D  Proofs for the critic

In this section, we collect together the statements and proofs of various technical results that underlie the critic's analysis in Section A.1. In Section D.1, we prove Lemma 1 that guarantees exactness of the critic on the induced MDP, whereas Section D.2 is devoted to proving our main guarantee for the critic, namely Proposition 2.

Let us introduce some additional notation that plays an important role in the proof. Recall the regression operator $\mathcal{R}_h^\pi$ and sup-norm projection operator $\mathcal{P}_h^\pi$ that were previously defined in equations (20a) and (20c), respectively. In addition to these two operators, our proof also makes use of the *approximation error operator*

$$\mathcal{A}_h^\pi(F)(s, a) \overset{def}{=} \langle \phi(s, a), \mathcal{P}_h^\pi(F) \rangle - (\mathcal{T}_h^\pi F)(s, a), \tag{36}$$

which is a mapping from $\mathcal{F}$ to itself.

## D.1 Proof of Lemma 1

By definition, the induced MDP differs from the original MDP only by the perturbation of the reward function. Thus, by definition of value functions, we can write

$$Q^\pi_{h,\hat{M}(\pi)}(s,a) - Q^\pi_h(s,a) = \sum_{\ell=h}^H \mathbb{E}_{(S_\ell, A_\ell) \sim \pi|(s,a)} \left[ \widehat{r}^\pi_h(S_\ell, A_\ell) - r_h(S_\ell, A_\ell) \right]. \tag{37a}$$

On the other hand, using the definition of $\underline{Q}^\pi_h$ and the Bellman conditions, we have

$$\begin{aligned}
\underline{Q}^\pi_h(s,a) - Q^\pi_h(s,a) &= \langle \phi(s,a), \underline{w}^\pi_h \rangle - \mathcal{T}^\pi_h(Q^\pi_{h+1})(s,a) \\
&= \left\{ \langle \phi(s,a), \underline{w}^\pi_h \rangle - \mathcal{T}^\pi_h(\underline{Q}^\pi_{h+1})(s,a) \right\} + \left\{ \mathcal{T}^\pi_h(\underline{Q}^\pi_{h+1})(s,a) + \mathcal{T}^\pi_h(Q^\pi_{h+1})(s,a) \right\} \\
&= \widehat{r}^\pi_h(s,a) - r_h(s,a) + \mathbb{E}_{S' \sim \mathbb{P}_h(s,a)} \mathbb{E}_{A' \sim \pi(\cdot|S')}(\underline{Q}^\pi_{h+1} - Q^\pi_{h+1})(S', A')
\end{aligned}$$

Applying this argument recursively to $\ell = h+1, \ldots, H$, we find that

$$\underline{Q}^\pi_h(s,a) - Q^\pi_h(s,a) = \sum_{\ell=h}^H \mathbb{E}_{(S_\ell, A_\ell) \sim \pi|(s,a)} \left[ \widehat{r}^\pi_h(S_\ell, A_\ell) - r_h(S_\ell, A_\ell) \right] \tag{37b}$$

Subtracting equation (37b) from equation (37a) yields the claim.

## D.2 Proof of Proposition 2

We split the proof into two parts, corresponding to the two bounds.

### D.2.1 Proof of the bound (23a)

We begin by proving the bound on the critic's estimate for the value function of the input policy $\pi$.

**High-level roadmap:** We begin by outlining the main steps in the proof. Our first step is to define a sequence of weight vectors $\widehat{w} \overset{def}{=} \{\widehat{w}^\pi_h\}_{h=1}^H$ such that

$$\left| \sum_{a_1 \in \mathcal{A}} \pi(a_1 \mid s_1) \langle \phi_1(s_1, a_1), \widehat{w}^\pi_1 \rangle - V^\pi_1(s_1) \right| \leq \sum_{h=1}^H \nu_h. \tag{38a}$$

Our second step is to show that conditioned on the good event $\mathcal{G}(\alpha)$ from equation (22), the sequence $\widehat{w}$ is feasible for the critic's convex program; this feasibility, combined with the optimality of $\underline{w}$, implies that

$$V^\pi_{1,\hat{M}(\pi)}(s_1) \overset{(i)}{=} \sum_{a_1 \in \mathcal{A}} \pi(a_1 \mid s_1) \langle \phi_1(s_1, a_1), \underline{w}^\pi_1 \rangle \leq \sum_{a_1 \in \mathcal{A}} \pi(a_1 \mid s_1) \langle \phi_1(s_1, a_1), \widehat{w}^\pi_1 \rangle. \tag{38b}$$

Here step (i) follows from Lemma 1, which guarantees that the estimated value functions $\underline{V}^\pi_h$ of the critic are exact in the induced MDP. Combining the two bounds (38a) and (38b) yields $V^\pi_{1,\hat{M}(\pi)}(s_1) \leq V^\pi_1(s_1) + \sum_{h=1}^H \nu_h$, as claimed in equation (23a).

It remains to prove our two auxiliary claims (38a) and (38b).

**Proof of claim (38a):** Given a policy $\pi$, we use backwards induction to define the sequence $\{\widehat{w}^\pi\}_{h=1}^H$ by first setting $\widehat{w}^\pi_{H+1} = 0$, and then defining

$$\widehat{w}^\pi_h \overset{def}{=} \mathcal{P}^\pi_h(\widehat{Q}^\pi_{h+1}) \qquad \text{for } h = H, H-1, \ldots, 1, \tag{39}$$

where $\widehat{Q}^\pi_{h+1}(s,a) \overset{def}{=} \langle \phi_{h+1}(s,a), \widehat{w}^\pi_{h+1} \rangle$. By construction, we have the bound $\|\widehat{w}^\pi_h\|_2 \leq \rho^w_h$ for all $h \in [H]$. The following lemma bounds the sup-norm distance between the induced linear $Q$-value function estimate, and the actual $Q^\pi$-value function.

**Lemma 3.** *The functions* $\{\widehat{Q}_h^\pi\}_{h=1}^H$ *defined by the best-predictor sequence* $\{\widehat{w}_h^\pi\}_{h=1}^H$ *from equation* (39) *satisfy the bound*

$$\left|\widehat{Q}_h^\pi(s,a) - Q_h^\pi(s,a)\right| \leq \sum_{\ell=h}^H \nu_\ell \qquad \text{for all } h \in [H]. \tag{40}$$

*Proof.* Introduce the shorthand $\Delta_h(s,a) \overset{def}{=} \widehat{Q}_h^\pi(s,a) - Q_h^\pi(s,a)$ for the error at stage $h$ to be bounded. Since $Q_h^\pi = \mathcal{T}_h^\pi(Q_{h+1}^\pi)$, we can write

$$\begin{aligned}
\Delta_h(s,a) &= \widehat{Q}_h^\pi(s,a) - Q_h^\pi(s,a) \\
&= \widehat{Q}_h^\pi(s,a) - (\mathcal{T}_h^\pi \widehat{Q}_{h+1}^\pi)(s,a) + (\mathcal{T}_h^\pi \widehat{Q}_{h+1}^\pi)(s,a) - \mathcal{T}_h^\pi(Q_{h+1}^\pi)(s,a) \\
&= \widehat{Q}_h^\pi(s,a) - (\mathcal{T}_h^\pi \widehat{Q}_{h+1}^\pi)(s,a) + \mathbb{E}_{S'\sim\mathbb{P}_h(s,a)}\mathbb{E}_{A'\sim\pi(\cdot|S')}\left[\widehat{Q}_{h+1}^\pi(S',A') - Q_{h+1}^\pi(S',A')\right] \\
&= \sum_{\ell=h}^H \mathbb{E}_{(S_\ell,A_\ell)\sim\pi|(s,a)}\left[\widehat{Q}_\ell^\pi(S_\ell,A_\ell) - \mathcal{T}_\ell^\pi(\widehat{Q}_{\ell+1}^\pi)(S_\ell,A_\ell)\right],
\end{aligned}$$

where the final equality follows by induction.

From the definition (39) of $\widehat{w}$ and the function estimate $\widehat{Q}_\ell^\pi(s,a) = \langle \phi_\ell(s,a), \widehat{w}_\ell^\pi \rangle$, combined with the Bellman approximation condition, we have

$$\left|\widehat{Q}_\ell^\pi(s,a) - (\mathcal{T}_\ell^\pi \widehat{Q}_{\ell+1}^\pi)(s,a)\right| \leq \mathcal{A}_\ell^\pi(\widehat{Q}_{\ell+1}^\pi) \leq \nu_\ell,$$

uniformly over all $\ell$, and over all state-action pairs $(s,a)$. Summing these bounds completes the proof. $\qquad\square$

**Proof of claim** (38b): In order to prove this claim, we need to exhibit a sequence $\xi = (\widehat{\xi}_1, \dots, \widehat{\xi}_H)$ such that the pair $(\widehat{\xi}, \widehat{w})$ are feasible for the critic's convex program (10). In particular, we need to ensure the following three conditions:

(a) $\|\widehat{w}_h^\pi\|_2 \leq \rho_h^w$ for all $h \in [H]$

(b) $\|\widehat{\xi}_h\|_{\Sigma_h} \leq \alpha_h$ for all $h \in [H]$.

(c) We have $\widehat{w}_h^\pi = \widehat{\xi}_h^\pi + \mathcal{R}_h^\pi(\widehat{Q}_{h+1}^\pi)$ for all $h \in [h]$.

Note that condition (a) is automatically satisfied by the definition (39) of $\widehat{w}$, since the projection $\mathcal{P}_h^\pi$ imposes this Euclidean norm bound.

It remains to exhibit a choice of $\widehat{\xi}$ such that conditions (b) and (c) hold. Since $\widehat{w}_h^\pi = \mathcal{P}_h^\pi(\widehat{Q}_h^\pi)$ by definition, condition (c) forces us to set

$$\widehat{\xi}_h^\pi = \mathcal{P}_h^\pi(\widehat{Q}_{h+1}^\pi) - \mathcal{R}_h^\pi(\widehat{Q}_{h+1}^\pi) = -\mathcal{E}_h^\pi(\widehat{Q}_{h+1}^\pi).$$

But since the event $\mathcal{G}(\alpha)$ holds by assumption, we have

$$\|\widehat{\xi}_h^\pi\|_{\Sigma_h} = \|\mathcal{E}_h^\pi(\widehat{Q}_{h+1}^\pi)\|_{\Sigma_h} \leq \alpha_h,$$

showing that this choice of $\widehat{\xi}$ satisfies condition (b).

### D.2.2  Proof of part (b)

Here we prove the bound (23b) stated in part (b) of the lemma, which provides an inequality on the value function error for an arbitrary policy.

Our proof is based on establishing an auxiliary result that implies the claim. In particular, we first show that for any policy $\widetilde{\pi}$, we have

$$\left|V_{1,\widehat{M}(\pi)}^{\widetilde{\pi}}(s_1) - V_1^{\widetilde{\pi}}(s_1)\right| \leq \sum_{h=1}^H \|\bar{\phi}_h^{\widetilde{\pi}}\|_{\Sigma_h^{-1}}\left\{\alpha_h + \|\mathcal{E}_h^\pi(\underline{Q}_{h+1}^\pi)\|_{\Sigma_h}\right\} + \sum_{h=1}^H \nu_h, \tag{41}$$

where $\bar{\phi}_h^{\widetilde{\pi}} \overset{def}{=} \mathbb{E}_{(S_h, A_h) \sim \widetilde{\pi}}[\phi(S_h, A_h)]$. Since $\|\mathcal{E}_h^\pi(Q_{h+1}^\pi)\|_{\Sigma_h} \leq \alpha_h$ conditioned on $\mathcal{G}(\alpha)$, this implies the claim.

Let us now prove the auxiliary claim (41). First, we observe that by definition, the perturbation in the reward can be written as

$$\widehat{r}_h^\pi(s, a) - r_h(s, a) \overset{(i)}{=} \langle \phi_h(s, a), \underline{w}_h^\pi \rangle - \mathcal{T}_h^\pi(\underline{Q}_{h+1}^\pi)(s, a)$$

$$\overset{(ii)}{=} \left\langle \phi_h(s, a), \underline{\xi}_h^\pi \right\rangle + \left\langle \phi_h(s, a), \mathcal{R}_h^\pi(\underline{Q}_{h+1}^\pi) \right\rangle - \mathcal{T}_h^\pi(\underline{Q}_{h+1}^\pi)(s, a)$$

$$\overset{(iii)}{=} \left\langle \phi_h(s, a), \underline{\xi}_h^\pi \right\rangle + \left\langle \phi_h(s, a), \mathcal{E}_h^\pi(\underline{Q}_{h+1}^\pi) \right\rangle + \mathcal{A}_h^\pi(\underline{Q}_{h+1}^\pi)(s, a),$$

where step (i) uses the definition $\underline{Q}_h^\pi(s, a) = \langle \phi_h(s, a), \underline{w}_h^\pi \rangle$; step (ii) uses the relation $\underline{w}_h^\pi = \underline{\xi}_h^\pi + \mathcal{R}_h^\pi(\underline{Q}_{h+1}^\pi)$; and step (iii) involves adding and subtracting $\left\langle \phi_h(s, a), \mathcal{P}_h^\pi(\underline{Q}_{h+1}^\pi) \right\rangle$, and using the definitions of the approximation error (36) and the error operator (21).

Since the induced MDP differs from the original only by the reward perturbation, we have

$$\left| V_{1, \hat{M}(\pi)}^{\widetilde{\pi}}(s_1) - V_1^{\widetilde{\pi}}(s_1) \right| = \left| \sum_{h=1}^H \mathbb{E}_{(S_h, A_h) \sim \widetilde{\pi}} \left[ \widehat{r}_h^\pi(S_h, A_h) - r_h(S_h, A_h) \right] \right|$$

$$= \left| \sum_{h=1}^H \mathbb{E}_{(S_h, A_h) \sim \widetilde{\pi}} \left[ \left\langle \phi_h(S_h, A_h), \underline{\xi}_h^\pi + \mathcal{E}_h^\pi(\underline{Q}_{h+1}^\pi) \right\rangle + \mathcal{A}_h^\pi(\underline{Q}_{h+1}^\pi)(S_h, A_h) \right] \right|.$$

We now observe that $|\mathcal{A}_h^\pi(\underline{Q}_{h+1}^\pi)(S_h, A_h)| \leq \nu_h$ by the Bellman closure assumption. As for the first term, introducing the shorthand $\bar{\phi}_h^{\widetilde{\pi}} \overset{def}{=} \mathbb{E}_{(S_h, A_h) \sim \widetilde{\pi}}[\phi_h(S_h, A_h)]$, we have

$$\mathbb{E}_{(S_h, A_h) \sim \widetilde{\pi}} \left[ \left\langle \phi_h(S_h, A_h), \underline{\xi}_h^\pi + \mathcal{E}_h^\pi(\underline{Q}_{h+1}^\pi) \right\rangle \right] \leq \|\bar{\phi}_h^{\widetilde{\pi}}\|_{\Sigma_h^{-1}} \|\underline{\xi}_h^\pi + \mathcal{E}_h^\pi(\underline{Q}_{h+1}^\pi)\|_{\Sigma_h}$$

$$\leq \|\bar{\phi}_h^{\widetilde{\pi}}\|_{\Sigma_h^{-1}} \left\{ \alpha_h + \|\mathcal{E}_h^\pi(\underline{Q}_{h+1}^\pi)\|_{\Sigma_h} \right\},$$

where the final step combines the triangle inequality, with the fact that $\|\underline{\xi}_h^\pi\|_{\Sigma_h} \leq \alpha_h$, since $\underline{\xi}_h^\pi$ must be feasible for the critic's convex program (10). Putting together the pieces yields the claim (41).

### D.3 Proof of Lemma 2

We now prove Lemma 2, which asserts that the good event $\mathcal{G}(\delta)$, as defined in equation (22), holds with high probability when the pessimism parameters are chosen according to equation (24).

Recall from equation (21) that for any pair $(Q, \pi)$, the associated parameter error is given by the difference $\mathcal{E}_h^\pi(Q) = \mathcal{R}_h^\pi(Q) - \mathcal{P}_h^\pi(Q)$. We begin with a simple lemma that decomposes this error into three terms. In order to state the lemma, we introduce two forms of error variables: statistical and approximation-theoretic.

Recall that $\mathcal{I}_h$ denotes the subset of indices associated with time step $h$. The first noise variables take the form

$$\eta_{hk}(Q, \pi) \overset{def}{=} r_{hk} + \mathbb{E}_{A' \sim \pi(\cdot|s_{hk})} Q(s_{h+1,k}, A') - (\mathcal{T}_h^\pi Q)(s_{hk}, a_{hk}), \tag{42a}$$

defined for each $h \in [H]$ and $k \in \mathcal{I}_h$. Note that conditionally on the pair $(s_{hk}, a_{hk})$, our sampling model and the definition of the Bellman operator $\mathcal{T}_h^\pi$ ensures that each $\eta_{hk}$ is zero-mean random variable, corresponding to a form of statistical error. Our analysis also involves some approximation error terms, in particular via the quantities

$$\Delta_{hk}(Q, \pi) \overset{def}{=} -\mathcal{A}_h^\pi(Q)(s_{hk}, a_{hk}) = (\mathcal{T}_h^\pi Q)(s_{hk}, a_{hk}) - \langle \phi_h(s_{hk}, a_{hk}), \mathcal{P}_h^\pi(Q) \rangle \tag{42b}$$

With these definitions, we have the following guarantee:

**Lemma 4** (Decomposition of $\mathcal{E}_h^\pi(Q)$)**.** *For any pair $(Q, \pi)$, we have the decomposition*

$$\mathcal{E}_h^\pi(Q) = e_h^\eta(Q, \pi) + e_h^\lambda(Q, \pi) + e_h^\Delta(Q, \pi), \tag{43}$$

*where the three error terms are given by*

$$e_h^\eta(Q, \pi) \overset{def}{=} \Sigma_h^{-1} \sum_{k \in \mathcal{I}_h} \phi_{hk} \eta_{hk}(Q, \pi), \qquad \text{(Statistical estimation error)} \qquad \text{(44a)}$$

$$e_h^\lambda(Q, \pi) \overset{def}{=} -\lambda \Sigma_h^{-1} \mathcal{P}_h^\pi(Q), \qquad \text{(Regularization error)}, \quad and \qquad \text{(44b)}$$

$$e_h^\Delta(Q, \pi) \overset{def}{=} \Sigma_h^{-1} \sum_{k \in \mathcal{I}_h} \phi_{hk} \Delta_{hk}(Q; \pi) \qquad \text{(Approximation error)}. \qquad \text{(44c)}$$

See Section D.3.1 for the proof of this claim.

The remainder of our analysis is focused on bounding these three terms. Analysis of the regularization error and approximation error terms is straightforward, whereas bounding the statistical estimation error requires more technical effort. We begin with the two easy terms.

**Regularization error:** Beginning with the definition (44b), we have

$$\|e_h^\lambda(Q, \pi)\|_{\Sigma_h} = \lambda \|\mathcal{P}_h^\pi(Q)\|_{\Sigma_h^{-1}} \overset{(i)}{\leq} \sqrt{\lambda} \|\mathcal{P}_h^\pi(Q)\|_2 \overset{(ii)}{\leq} \sqrt{\lambda}, \qquad (45)$$

where step (i) follows since $\Sigma_h \succeq \lambda I$; and inequality (ii) follows from the bound $\|\mathcal{P}_h^\pi(Q)\|_2 \leq \rho_h^w \leq 1$, guaranteed by the definition of $\mathcal{P}_h^\pi$.

**Approximation error:** By definition, we have $\|e_h^\Delta(Q, \pi)\|_{\Sigma_h} = \|\sum_{k \in \mathcal{I}_h} \phi_{hk} \Delta_{hk}(Q, \pi)\|_{\Sigma_h^{-1}}$. By the Bellman approximation condition, we have $|\Delta_{hk}(Q, \pi)| \leq \nu_h$ uniformly over all $k$. Consequently, applying Lemma 8 (Projection Bound) from the paper Zanette et al. (2020b) guarantees that

$$\|e_h^\Delta(Q, \pi)\|_{\Sigma_h} \leq \sqrt{n_h} \nu_h. \qquad (46)$$

**Statistical estimation error:** Lastly, we turn to the analysis of the statistical estimation error. In particular, we prove the following guarantee:

**Lemma 5.** *There is a universal constant $c > 0$ such that*

$$\|e_h^\eta(Q, \pi)\|_{\Sigma_h}^2 \leq c \left\{ 1 + d_h \log \left( 1 + \frac{T}{d_h \lambda} \right) + d_h \log \left( 1 + 8\sqrt{T} \right) + d_h \log \left( 1 + 16R\sqrt{T} \right) + \log \frac{H}{\delta} \right\} \qquad (47)$$

*uniformly over all $Q \in \mathcal{Q}_h$, $\pi \in \Pi_{soft}(R)$ and $h \in [H]$ with probability at least $1 - \delta$.*

See Section D.3.2 for the proof of this claim.

**Putting together the pieces:** By combining our three bounds—namely, equations (45), (46) and (47), we conclude that with the choice

$$\alpha_h(\delta) \overset{def}{=} \sqrt{\lambda} + \sqrt{n_h} \nu_h +$$
$$c \left\{ 1 + d_h \log \left( 1 + \frac{T}{d_h \lambda} \right) + d_h \log \left( 1 + 8\sqrt{T} \right) + d_h \log \left( 1 + 16R\sqrt{T} \right) + \log \frac{H}{\delta} \right\}^{1/2},$$

the good event $\mathcal{G}(\delta)$ holds with probability at least $1 - \delta$. This completes the proof of Lemma 2.

It remains to prove the two auxiliary lemmas that we stated: namely, Lemma 4 that gave a decomposition of the parameter error, and Lemma 5 that bounded the statistical error. We do so in Sections D.3.1 and D.3.2, respectively.

### D.3.1 Proof of Lemma 4

Starting with the definition (20a) of the regression operator $\mathcal{R}_h^\pi$, we have

$$\mathcal{R}_h^\pi(Q) \stackrel{def}{=} \Sigma_h^{-1} \sum_{k \in \mathcal{I}_h} \phi_{hk}[r_{hk} + \mathbb{E}_{A' \sim \pi(\cdot|s_{hk})} Q(s_{h+1,k}, A')]$$

$$\stackrel{(i)}{=} \Sigma_h^{-1} \sum_{k \in \mathcal{I}_h} \phi_{hk}[(\mathcal{T}_h^\pi Q)(s_{hk}, a_{hk})] + \underbrace{\Sigma_h^{-1} \sum_{k \in \mathcal{I}_h} \phi_{hk} \eta_{hk}(Q, \pi)}_{=e_h^\eta(Q,\pi)}$$

where equality (i) follows by adding and subtracting terms, and using the definition (42a) of $\eta_{hk}$.

Next we use the definition (42b) of the approximation error terms $\Delta_{hk}$ to find that

$$\mathcal{R}_h^\pi(Q) = \xi_h + \Sigma_h^{-1}\left(\sum_{k \in \mathcal{I}_h} \phi_{hk}\big[\langle \phi_{hk}, \mathcal{P}_h^\pi(Q)\rangle + \Delta_{hk}(Q,\pi)\big]\right) + e_h^\eta(Q,\pi)$$

Since $\Sigma_h = \sum_{k \in \mathcal{I}_h} \phi_{hk}\phi_{hk}^\top + \lambda I$, we can write

$$w_h(Q, \pi, \xi_h) = \xi_h + \Sigma_h^{-1}\Big\{\Sigma_h w_h^\star(Q,\pi) + \sum_{k \in \mathcal{I}_h} \phi_{hk}\Delta_{hk}(Q,\pi) - \lambda w_h^\star(Q,\pi)\Big\} + e_h^\eta$$

$$= \xi_h + w_h^\star(Q,\pi) + \Sigma_h^{-1}\left(\sum_{k \in \mathcal{I}_h} \phi_{hk}\Delta_{hk}(Q,\pi) - \lambda w_h^\star(Q,\pi)\right) + e_h^\eta$$

$$= \xi_h + w_h^\star(Q,\pi) + e_h^\eta + e_h^\lambda + e_h^\Delta,$$

which completes the proof.

### D.3.2 Proof of Lemma 5

From the definition (3a), we need to study the constrained class of linear action-value functions based on radii $\rho_h^w \in (0,1]$ for all $h \in [H]$. As for the constraint defining the soft-max policy class (3b), let us upper bound how large the $\ell_2$-norm of the actor's parameter vector can be over $T$ iterations.

Based on the actor's updates, we have the bound

$$\|\theta_{t,h}\|_2 = \|\sum_{t=1}^T \eta w_{t,h}\|_2 \leq \eta \sum_{t=1}^T \|w_{t,h}\|_2 \stackrel{(i)}{\leq} \eta T \rho_h^w \stackrel{(ii)}{\leq} \eta T,$$

where step (i) follows from the definition of the critic's program (10), and step (ii) follows from the assumption $\rho_h^w \in (0,1]$. Thus, we are assured that $R = \eta T$ is an upper bound on this $\ell_2$-norm.

We make use of a discretization argument to control the associated empirical process. Let $N_\infty(\epsilon; \mathcal{Q})$ denote the cardinality of the smallest $\epsilon$-covering of $\mathcal{Q}$ in the sup-norm—that is, a collection $\{Q^i\}_{i=1}^N$ such that for all $Q \in \mathcal{Q}$, we can find some $i \in [N]$ such that

$$\|Q - Q^i\|_\infty = \sup_{(s,a)} |Q(s,a) - Q^i(s,a)| \leq \epsilon.$$

Similarly, we let $N_{\infty,1}(\epsilon; \Pi(R))$ denote an $\epsilon$-cover of $\Pi(R)$ when measuring distances with the norm

$$\|\pi - \pi'\|_{\infty,1} \stackrel{def}{=} \sup_s \sum_{a \in \mathcal{A}} |\pi(a \mid s) - \pi'(a \mid s)|. \tag{48}$$

We have the following bounds on these covering numbers:

**Lemma 6** (Covering number bounds). *For any $\epsilon \in (0,1)$, we have*

$$\log N_\infty(\epsilon; \mathcal{Q}) \leq d \log\left(1 + \tfrac{2}{\epsilon}\right) \qquad and \tag{49a}$$

$$\log N_{\infty,1}(\epsilon; \Pi(R)) \leq d \log\left(1 + \tfrac{16R}{\epsilon}\right). \tag{49b}$$

See Section D.3.3 for the proofs of these claims.

For any $\epsilon \in (0, 1)$, we define

$$\beta(\epsilon) \stackrel{def}{=} d \log\left(1 + \tfrac{T}{d\lambda}\right) + \log N_\infty(\epsilon; \mathcal{Q}) + \log N_{\infty,1}(\epsilon; \Pi_{soft}) + \log \frac{H}{\delta} \tag{50}$$

Given this definition and the bounds from Lemma 6, the proof of Lemma 5 is reduced to showing that for any $\epsilon \in (0, 1)$, there is a universal constant $c$ such that

$$\max_{h \in [H]} \sup_{\substack{Q \in \mathcal{Q}_h \\ \pi \in \Pi_{soft}}} \|e_h^\eta(Q, \pi)\|_{\Sigma_h} \leq c\sqrt{\beta(\epsilon)} + 4\sqrt{T}\epsilon \tag{51}$$

with probability at least $1 - \delta$. The claim stated in Lemma 5 follows from the choice $\epsilon = \frac{1}{4\sqrt{T}}$. The remainder of our proof is devoted to the proof of this claim.

**Proof of the claim** (51): Let us recall the definition

$$\eta_{hk}(Q, \pi) = r_{hk} + \mathbb{E}_{A' \sim \pi_h(\cdot | s_{hk})} Q(s_{h+1,k}, A') - (\mathcal{T}_h^\pi Q)(s_{hk}, a_{hk}).$$

Consequently, by starting with the definition of $e_h^\eta$ and applying the triangle inequality, we obtain the upper bound $\|e_h^\eta(Q, \pi)\|_{\Sigma_h} = \|\sum_{k \in \mathcal{I}_h} \phi_{hk} \eta_{hk}(Q, \pi)\|_{\Sigma_h^{-1}} \leq Z_1 + Z_2(Q, \pi)$, where

$$Z_1 \stackrel{def}{=} \|\sum_{k \in \mathcal{I}_h} \phi_{hk} \underbrace{[r_{hk} - r(s_{hk}, a_{hk})]}_{\stackrel{def}{=} Y_{hk}}\|_{\Sigma_h^{-1}} \text{ and}$$

$$Z_2(Q, \pi) \stackrel{def}{=} \left\|\sum_{k \in \mathcal{I}_h} \phi_{hk}[Q(s_{h+1,k}, \pi) - \mathbb{E}_{S' \sim \mathbb{P}(\cdot | s_{hk}, a_{hk})} Q(S', \pi)]\right\|_{\Sigma_h^{-1}}$$

For a fixed $(\pi, Q)$ and conditioned on the sampling history, both $Z_1$ and $Z_2$ are mean zero. Note that $Z_1$ is independent of the pair $(Q, \pi)$, so that its analysis does not require discretization techniques. On the other hand, analyzing $Z_2(Q, \pi)$ does require a reduction step via discretization, with which we begin.

Introducing the shorthand $N = N(\epsilon, \mathcal{Q})$, let $\{Q^i\}_{i=1}^N$ be an $\epsilon$-cover of the set $\mathcal{Q}$ in the sup-norm. Similarly, with the shorthand $J = N(\epsilon, \Pi)$, let $\{\pi^j\}_{j=1}^J$ be an $\epsilon$-cover of $\Pi$ in the norm (48). For a given $Q$, let $Q^i$ denote the member of the cover such that $\|Q - Q^i\|_\infty \leq \epsilon$. With this choice, we have

$$Z_2(Q, \pi) = Z_2(Q^i, \pi) + \{Z_2(Q, \pi) - Z_2(Q^i, \pi)\}.$$

Similarly, let $\pi^m$ be a member of the cover such that $\|\pi(\cdot \mid s) - \pi^m(\cdot \mid s)\|_1 \leq \epsilon$ for all $s$. With this choice, we have

$$Z_2(Q, \pi) \leq Z_2(Q^i, \pi^m) + \underbrace{\{Z_2(Q^i, \pi) - Z_2(Q^i, \pi^m)\}}_{D^\pi} + \underbrace{\{Z_2(Q, \pi) - Z_2(Q^i, \pi)\}}_{D^Q}.$$

We begin by bounding the two discretization errors. By the triangle inequality, we have

$$D^Q \leq \left\|\sum_{k \in \mathcal{I}_h} \phi_{hk}\underbrace{[Q(s_{h+1,k}, \pi) - Q^i(s_{h+1,k}, \pi) + \mathbb{E}_{S' \sim p(s_{hk}, a_{hk})}(Q(S', \pi) - Q^i(S', \pi))]}_{\stackrel{def}{=} E_{hk}^i(Q, \pi)}\right\|_{\Sigma_h^{-1}}.$$

Our choice of discretization ensures that $|E_{hk}^i(Q, \pi)| \leq 2\epsilon$ uniformly for all $(h, k)$ and $(Q, \pi)$. Applying Lemma 8 (Projection Bound) from the paper Zanette et al. (2020b) ensures that $D^Q \leq 2\epsilon\sqrt{T}$. To be clear, this is a deterministic claim; it holds uniformly over the choices of $Q$, $Q^i$, and $\pi$. A similar argument yields that $D^\pi \leq 2\epsilon\sqrt{T}$.

Putting togther the pieces yields that for any $(Q, \pi)$, we have the bound

$$Z_2(Q, \pi) \leq \max_{\substack{i \in [N] \\ j \in [M]}} Z_2(Q^i, \pi^j) + 4\sqrt{T}\epsilon. \tag{52}$$

We now need to bound $Z_1$ along with $Z_2(Q^i, \pi^j)$ for a fixed pair $(Q^i, \pi^j)$. In order to do so, we apply known self-normalized tail bounds de la Pena et al. (2009), which apply to sums of the form $\|\sum_{k \in \mathcal{I}_h} \phi_{hk} V_{hk}\|_{\Sigma_h^{-1}}$, where the $V_{hk}$ form a martingale difference sequence with conditionally sub-Gaussian tails. Note that $Z_1$ is of this general form with $V_{hk} = Y_{hk}$, which is a 1-sub-Gaussian variable by assumption. On the other hand, the variable $Z_2(Q^i, \pi^j)$ is of this form with

$$V_{hk} = Q^i(s_{h+1,k}, \pi^j) - \mathbb{E}_{S' \sim p(s_{hk}, a_{hk})} Q^i(S', \pi^j).$$

Since $|V_{hk}| \leq 1$ due to the uniform boundedness of $Q^i$, this is a 1-sub-Gaussian variable as well.

Consequently, Theorem 1 from the paper Abbasi-Yadkori et al. (2011) ensures that

$$\mathbb{P}\left(\max\{Z_1, Z_2(Q^i, \pi^j)\} \geq \log \frac{\det \Sigma_h}{\det \lambda I} + 2 \log \frac{1}{\delta}\right) \leq \delta.$$

Note that $\det \lambda I = \lambda^{d_h}$. Moreover, Lemma 10 (Determinant-Trace Inequality) in Abbasi-Yadkori et al. (2011) yields $\log \det \Sigma_h \leq d_h \log\left(\lambda + \frac{T}{d_h}\right)$.

Putting together the pieces, taking a union bound over the two covers yields that, for each fixed $h \in [H]$, we have

$$\|e_h^\eta(Q, \pi)\|_{\Sigma_h^{-1}} \leq d_h \log\left(1 + \frac{T}{d_h \lambda}\right) + \log N_\infty(\epsilon; \mathcal{Q}) + \log N_{\infty,1}(\epsilon; \Pi) + \log\left(\frac{1}{\delta}\right) + 4\sqrt{T}\epsilon$$

with probability at least $1 - \delta$. Finally, we take a union bound over all $h \in [H]$, which forces us to redefine $\delta$ to $\frac{\delta}{H}$ in the above bound. This completes the proof of the uniform bound (51).

### D.3.3 Proof of Lemma 6

Since $\|\phi(s,a)\|_2 \leq 1$, for any pair of weight vectors $w, w' \in \mathbb{R}^d$, we have $\sup_{(s,a)} |\langle \phi(s,a), w - w' \rangle \|_2 \leq \|w - w'\|_2$. Thus, the bound (49a) follows from standard results on coverings of Euclidean balls (cf. Example 5.8 in the book Wainwright (2019)).

As for the bound (49b), we claim that

$$\sum_{a \in \mathcal{A}} |\pi_{\theta'}(a \mid s) - \pi_\theta(a \mid s)| \leq 8\|\theta - \theta'\|_2, \qquad \text{for all } s \in \mathcal{S}. \tag{53}$$

Taking this claim as given for the moment, it suffices to obtain an $\epsilon/8$-cover of the ball $\mathcal{B}(R)$ in the $\ell_2$-norm, and applying the same standard results yields the claimed bound (49b).

It remains to prove the claim (53).

**Proof of the claim** (53): Let us state and prove the claim (53) more formally as a lemma. It applies to the softmax policy $\pi_\theta(a \mid s) = \frac{\exp\{\langle \phi(s,a), \theta\rangle\}}{\sum_{a' \in \mathcal{A}} \exp(\langle \phi(s,a'), \theta\rangle)}$.

**Lemma 7** (Nearby Policies). *Consider a feature mapping $\phi : \mathcal{S} \times \mathcal{A} \to \mathbb{R}^d$ such that $\|\phi(s,a)\|_2 \leq 1$ uniformly for all pairs $(s,a)$. Then for all $s \in \mathcal{S}$, we have*

$$\sum_{a \in \mathcal{A}} |\pi_{\theta'}(a \mid s) - \pi_\theta(a \mid s)| \leq 8\|\theta - \theta'\|_2, \tag{54}$$

*valid for any pair $\theta, \theta' \in \mathbb{R}^d$ such that $\|\theta - \theta'\|_2 \leq \frac{1}{2}$.*

*Proof.* Dividing $\pi_{\theta'}(s,a)$ by $\pi_\theta(s,a)$ yields

$$\begin{aligned}
T \stackrel{def}{=} \frac{\pi_{\theta'}(a \mid s)}{\pi_\theta(a \mid s)} &= \frac{e^{\langle \phi(s,a), \theta'\rangle}}{e^{\langle \phi(s,a), \theta\rangle}} \times \frac{\sum_{a''} e^{\langle \phi(s,a''), \theta\rangle}}{\sum_{\tilde{a}} e^{\langle \phi(s,\tilde{a}), \theta'\rangle}} \\
&= e^{\langle \phi(s,a), \theta' - \theta\rangle} \times \sum_{a''} \left(e^{\langle \phi(s,a''), \theta - \theta'\rangle} \times \frac{e^{\langle \phi(s,a''), \theta'\rangle}}{\sum_{\tilde{a}} e^{\langle \phi(s,\tilde{a}), \theta'\rangle}}\right) \\
&= e^{\langle \phi(s,a), \theta' - \theta\rangle} \times \sum_{a''} \pi_{\theta'}(a'' \mid s) e^{\langle \phi(s,a''), \theta - \theta'\rangle}.
\end{aligned}$$

By Cauchy-Schwarz and the assumption on $\phi$, we have the bound $|\langle \theta(s,a), \gamma\rangle| \leq \|\gamma\|_2$, valid for any vector $\gamma$. Monotonicity of the exponential allows us to exponentiate this inequality. Combined with the fact that $\pi_{\theta'}(a'' \mid s) \geq 0$, we find that

$$T \leq e^{\|\theta'-\theta\|_2} \sum_{a''\in\mathcal{A}} \pi_{\theta'}(a'' \mid s) e^{\|\theta-\theta'\|_2} \stackrel{(i)}{=} e^{2\|\theta-\theta'\|_2} \stackrel{(ii)}{\leq} 1 + 4\|\theta - \theta'\|_2, \qquad (55)$$

where step (i) uses the fact that $\pi_\theta$ is a probability distribution over the action space; and step (ii) follows by combining the elementary inequality $e^x \leq 1 + 2x$, valid for all $x \in [0,1]$, with our assumption that $\|\theta - \theta'\|_2 \leq 1/2$.

Recalling that $T = \frac{\pi_{\theta'}(a|s)}{\pi_\theta(a|s)}$, re-arranging the inequality (55) yields the bound

$$\pi_{\theta'}(a \mid s) - \pi_\theta(a \mid s) \leq 4\pi_\theta(a \mid s)\,\|\theta - \theta'\|_2,$$

valid uniformly over all pairs $(s,a)$. We can apply the same argument with the roles of $\theta$ and $\theta'$ reversed, and combining the two bounds yields

$$|\pi_{\theta'}(a \mid s) - \pi_\theta(a \mid s)| \leq 4\|\theta - \theta'\|_2 \max\{\pi_\theta(a \mid s),\, \pi_{\theta'}(a \mid s)\},$$

again uniformly over all pairs $(s,a)$. Now summing over the actions $a$, we find that

$$\sum_{a\in\mathcal{A}} |\pi_{\theta'}(a \mid s) - \pi_\theta(a \mid s)| \leq 4 \sum_{a\in\mathcal{A}} \max\{\pi_\theta(a \mid s), \pi_{\theta'}(a \mid s)\}\,\|\theta - \theta'\|_2$$

$$\leq 4 \sum_{a\in\mathcal{A}} \{\pi_\theta(a \mid s) + \pi_{\theta'}(a \mid s)\}\|\theta - \theta'\|_2$$

$$= 8\|\theta - \theta'\|_2,$$

where the last step uses the fact that $\pi_\theta$ and $\pi_{\theta'}$ are probability distributions over the action space. Note that this inequality holds for all states $s$, as claimed. $\qquad\square$

# E   Actor's analysis: Proof of Proposition 3

In order to prove this claim, we require an auxiliary result that re-expresses the mirror update rule. Given the $Q$-value function $Q(s,a) \stackrel{def}{=} \langle \phi(s,a), w\rangle$, consider the linear update $\theta^+ \stackrel{def}{=} \theta + \eta w$, and the induced soft-max policy $\pi_{\theta^+}$. The following auxiliary result extracts a useful property of this update:

**Lemma 8** (Update in Natural Policy Gradient). *For any function $F : \mathcal{S} \to \mathbb{R}$, we have*

$$Q(s,a) - F(s) = \frac{1}{\eta}\left[\log\frac{\pi_{\theta^+}(s,a)}{\pi_\theta(s,a)} + \log\left(\sum_{a'\in\mathcal{A}} \pi_\theta(s,a') e^{\eta\left(Q(s,a')-F(s)\right)}\right)\right], \qquad (56)$$

*valid for all pairs $(s,a)$.*

See Section E.1 for the proof of this claim.

Turning to the proof of the proposition, we have

$$V^\pi_{1,M_t}(s_1) - V^{\pi_t}_{1,M_t}(s_1) \stackrel{(i)}{=} \sum_{h=1}^H \mathbb{E}_{(S_h,A_h)\sim\pi}\left[G^{\pi_t}_{h,M_t}(S_h, A_h)\right] \stackrel{(ii)}{=} \frac{1}{\eta}\sum_{h=1}^H X_{h,t}, \qquad (57)$$

where we have introduced the shorthand

$$X_{h,t} \stackrel{def}{=} \mathbb{E}_{(S_h,A_h)\sim\pi}\left[\log\frac{\pi_{\theta_{t+1}}(S_h, A_h)}{\pi_{\theta_t}(S_h, A_h)} + \log\left(\mathbb{E}_{A'_h\sim\pi_t(\cdot|S_h)}\left[e^{\eta G^{\pi_t}_{h,M_t}(S_h,A'_h)}\right]\right)\right]. \qquad (58)$$

Here step (i) follows from the simulation lemma (e.g., Kakade et al. (2003)), and step (ii) makes use of Lemma 8 with $F(s) = V^{\pi_t}_{h,M_t}(s)$, along with the definition of the advantage function—namely, $G^{\pi_t}_{h,M_t}(s,a) = Q^{\pi_t}_{h,M_t}(s,a) - V^{\pi_t}_{h,M_t}(s)$.

For each $h \in [H]$ and $t \in [T]$, we now bound the two terms within the definition (58) of $X_{h,t}$ separately. In particular, we derive a telescoping relationship for the first term, and a uniform bound on the second term.

**First term:** For any pair of policies $\pi, \widetilde{\pi}$ and $s$, we introduce the shorthand

$$D_s(\pi; \widetilde{\pi}) \overset{def}{=} KL\left(\pi(\cdot \mid s) \| \widetilde{\pi}(\cdot \mid s)\right).$$

From the definition of KL divergence, for each $s_h$, we have

$$\sum_{a_h \in \mathcal{A}} \pi(a_h \mid s_h) \log \frac{\pi_{t+1}(s_h, a_h)}{\pi_t(s_h, a_h)} = \sum_{a_h} \pi(a_h \mid s_h) \left[ \log \frac{\pi_{t+1}(s_h, a_h)}{\pi(s_h, a_h)} - \log \frac{\pi_t(s_h, a_h)}{\pi(s_h, a_h)} \right]$$

$$= -D_{s_h}(\pi; \pi_{t+1}) + D_{s_h}(\pi; \pi_t). \tag{59}$$

**Second term:** We begin with the elementary inequality $e^x \leq 1 + x + x^2$ valid for all $x \in [0, 1]$. By assumption, we have $|\eta G^{\pi_t}_{h, M_t}(s, a)| \leq 2\eta \leq 1$ for any pair $(s, a)$, and hence

$$e^{\eta G^{\pi_t}_{h, M_t}(s,a)} \leq 1 + \left(\eta G^{\pi_t}_{h, M_t}(s, a)\right) + \left(\eta G^{\pi_t}_{h, M_t}(s, a)\right)^2 \leq 1 + \left(\eta G^{\pi_t}_{h, M_t}(s, a)\right) + 4\eta^2.$$

By definition of the advantage function, we have $\mathbb{E}_{A'_h \sim \pi_t}\left[G^{\pi_t}_{h, M_t}(s_h, A'_h)\right] = 0$, so that we have

$$\log\left(\mathbb{E}_{A'_h \sim \pi_t} e^{\eta G^{\pi_t}_{h, M_t}(s_h, A'_h)}\right) \leq \log\left(1 + 4\eta^2\right) \leq 4\eta^2. \tag{60}$$

**Combining the pieces:** Combining the bounds (59) and (60) yields

$$\frac{1}{\eta} X_{h,t} \leq \frac{1}{\eta} \mathbb{E}_{(S_h) \sim \pi}\left[-D_{S_h}(\pi; \pi_{t+1}) + D_{S_h}(\pi; \pi_t)\right] + 4\eta.$$

Averaging this bound over all $t \in [T]$ and exploiting the telescoping of the terms yields

$$\frac{1}{\eta T} \sum_{t=1}^{T} X_{h,t} \leq \frac{1}{\eta T} \mathbb{E}_{S_h \sim \pi}\left[-D_{S_h}(\pi; \pi_{t+1}) + D_{S_h}(\pi; \pi_1)\right] + 4\eta$$

$$\overset{(i)}{\leq} \frac{1}{\eta T} \mathbb{E}_{(S_h) \sim \pi} D_{S_h}(\pi; \pi_1) + 4\eta$$

$$\overset{(ii)}{\leq} \frac{1}{\eta T} \log(|\mathcal{A}|) + 4\eta,$$

where step (i) follows by non-negativity of the KL divergence; and step (ii) uses the fact that the KL divergence is at most $\log(|\mathcal{A}|)$. Summing these bounds over $h \in [H]$ yields

$$\frac{1}{T} \sum_{t=1}^{T} \left\{ V^{\pi}_{1, M_t}(s_1) - V^{\pi_t}_{1, M_t}(s_1) \right\} = \frac{1}{\eta T} \sum_{t=1}^{T} \sum_{h=1}^{H} X_{h,t} \leq H \left\{ \frac{1}{\eta T} \log(|\mathcal{A}|) + 4\eta \right\},$$

thereby establishing the claim (27a).

Finally, the bound (27b) follows by making the particular stepsize choice $\eta = \sqrt{\frac{\log |\mathcal{A}|}{T}}$. Note that the assumed lower bound $T \geq \log |\mathcal{A}|$ ensures that $\eta \leq 1$, as required to apply the bound (27a).

### E.1 Proof of Lemma 8

By definition of the soft-max policy, we have $\pi_{\theta^+}(s, a) = \frac{\exp(\langle \phi(s,a), \theta^+ \rangle)}{\sum_{a' \in \mathcal{A}} e^{\langle \phi(s,a'), \theta^+ \rangle}}$. Since $\theta_+ = \theta + \eta w$, we can write

$$\pi_{\theta^+}(s, a) = \frac{e^{\langle \phi(s,a), \theta + \eta w \rangle}}{\sum_{a' \in \mathcal{A}} e^{\langle \phi(s,a'), \theta + \eta w \rangle}} = \frac{e^{\langle \phi(s,a), \theta \rangle} e^{\eta \langle \phi(s,a), w \rangle}}{\sum_{a' \in \mathcal{A}} e^{\langle \phi(s,a'), \theta \rangle} e^{\eta \langle \phi(s,a'), w \rangle}}$$

$$= \frac{e^{\langle \phi(s,a), \theta \rangle}}{\sum_{\tilde{a} \in \mathcal{A}} e^{\langle \phi(s,\tilde{a}), \theta \rangle}} \times \frac{e^{\eta \langle \phi(s,a), w \rangle}}{\sum_{a' \in \mathcal{A}} \frac{e^{\langle \phi(s,a'), \theta \rangle}}{\sum_{\tilde{a} \in \mathcal{A}} e^{\langle \phi(s,\tilde{a}), \theta \rangle}} e^{\eta \langle \phi(s,a'), w \rangle}}$$

$$= \pi_\theta(s, a) \times \frac{e^{\eta \langle \phi(s,a), w \rangle}}{\sum_{a' \in \mathcal{A}} \pi_\theta(s, a') e^{\eta \langle \phi(s,a'), w \rangle}}$$

$$= \pi_\theta(s, a) \times \frac{e^{\eta Q(s,a)}}{\sum_{a' \in \mathcal{A}} \pi_\theta(s, a') e^{\eta Q(s,a')}}$$

where the last step uses the definition of $Q$. Multiplying both sides by $e^{-F(s)}$ and re-arranging yields

$$\frac{\pi_{\theta^+}(s,a)}{\pi_\theta(s,a)} \sum_{a' \in \mathcal{A}} \pi_\theta(s,a') e^{\eta[Q(s,a')-F(s)]} = e^{\eta[Q(s,a)-F(s)]},$$

which is equivalent to the claim.

## F   Proof of Theorem 2

We now turn to the proof of the lower bound stated in Theorem 2. In Section F.1, we describe the class of MDPs used in the construction, along with the data generating procedure. Section F.2 provides the core argument, which involves three auxiliary lemmas. These lemmas are proved in Sections F.3, F.4 and F.5, respectively.

### F.1   MDP class and data collection

For a given horizon $H$ and dimension $d$, we define a family of MDPs that are parameterized by a Boolean vector $u = (u_1, \ldots, u_H) \in \{-1, +1\}^{dH}$, where each $u_h \in \{-1, +1\}^d$. For a given Boolean vector $u$, the associated MDP $M_u$ has the following structure:

**State space and transition:** At each time step $h$, there is only one state—viz. $\mathcal{S} = \{s\}$. Since there is a single state, the transition is deterministic into the same state.

**Action space:** At each time step $h$, the action space is given by $\mathcal{A} = \{-1, 0, +1\}^d$.

**Feature map:** At each time step $h$, the feature map $\phi : \mathcal{S} \times \mathcal{A} \to \mathbb{R}^{d+1}$ takes the form

$$\phi(s,a) = \left[ \frac{a}{\sqrt{2d}}, \frac{1}{\sqrt{2}} \right]. \tag{61}$$

Notice that by construction, we have the bound $\|\phi(s,a)\|_2 = \sqrt{\frac{\|a\|_2^2}{2d} + \frac{1}{2}} \le 1$ for any state-action pair.

**Reward mean:** The mean reward at time step $h$ is proportional to the inner product $\langle a, u_h \rangle$, where $u_h \in \{-1, 1\}^d$ is the sub-vector associated with time step $h$. More precisely, we have

$$r_h(s,a) = \langle \phi_h(s,a), [\delta u_h \quad 0] \rangle = \frac{\delta}{2\sqrt{d}} \langle a, u_h \rangle, \tag{62}$$

where $\delta > 0$ is a parameter to be specified in the proof.

**Low-rank MDP model:** It is easy to verify that the MDP so defined is low-rank; here we only verify explicitly the regularity conditions about the size of the radii so that the setting for the lower bound matches the setting that PACLE can handle. We need to verify explicitly that we can represent the action value function for any policy $\pi$, namely that there exists $w_h^\pi$ such that the action value function $Q_h^\pi(s,a) = \langle \phi_h(s,a), w_h^\pi \rangle$ with $\|w_h^\pi\|_2 \le (H - h + 1)/(2H)$. One can verify that for any policy $\pi$ we have

$$w_h^\pi = [\delta u_h, \sqrt{2} V_{h+1}^\pi], \qquad \forall h \in [H]. \tag{63}$$

A sufficient condition for the regularity conditions to be satisfied is when

$$\delta \|u_h\|_2 \le 1/(2H) \to \delta \le \frac{1}{2\sqrt{d}H}, \tag{64}$$

which implies $|V_h^\pi| \le (H - h + 1)/(2H)$ and hence

$$\|w_h^\pi\|_2 \le \delta_2 \|u_h\|_2 + \sqrt{2}|V_{h+1}^\pi| \le (H - h + 1)/(2H), \qquad \forall h \in [H]. \tag{65}$$

In Lemma 10 we choose $\delta = \frac{d\sqrt{H}}{\sqrt{2n}}$ which implies the lemma holds when

$$\frac{d\sqrt{H}}{\sqrt{2n}} \le \frac{1}{2\sqrt{d}H} \to n \ge 2d^3 H^3. \tag{66}$$

**Reward observations:** We observe the mean reward contaminated by additive Gaussian noise, so that the reward distribution has the form

$$R_h(s,a) \sim \mathcal{N}\left( \frac{\delta}{\sqrt{2d}} \langle a, u_h \rangle, 1 \right). \tag{67}$$

**Data collection:** We assume that the $n$ samples are collected according to the following non-adaptive process.

- Each time step $h \in [H]$ is allocated $n_H \overset{def}{=} n/H$ samples (assumed to be an integer for simplicity).
- For each $h$, the dataset $\mathcal{D}_h$ is generated by playing each action $a \in \{e_1, \ldots, e_d, 0\}$ exactly $n_H/(d+1)$ times, where $e_j \in \{0,1\}^d$ denotes the standard basis vector with a single one in index $j$.

### F.2 Main argument

With this set-up, we now introduce the three lemmas that form the core of the proof. For any given $u \in \{-1, +1\}^{dH}$, let $\mathbb{Q}_u$ denote the distribution of the data $\mathcal{D}$ when the sampling process is applied to the MDP $M_u$, and let $\mathbb{E}_u$ denote expectations under this distribution. Our first lemma exploits the Assouad construction so as to reduce the problem of finding a good policy to a family of testing problems.

**Lemma 9** (Reduction to testing). *For any estimated policy $\pi_{\mathrm{ALG}}$, we have*

$$\sup_{u \in \mathcal{U}} \mathbb{E}_u[V_u^\star - V_u^{\pi_{\mathrm{ALG}}}] \geq \frac{\delta}{\sqrt{2d}} \frac{dH}{2} \min_{\substack{u,u' \in \mathcal{U} \\ D_H(u;u')=1}} \inf_\psi \left[ \mathbb{Q}_u(\psi(\mathcal{D}) \neq u) + \mathbb{Q}_{u'}(\psi(\mathcal{D}) \neq u') \right], \quad (68)$$

*where a test function $\psi$ is a measurable function of the data taking values in $\{u, u'\}$.*

See Section F.3 for the proof.

Our second lemma involves further lower bounding the testing error in the bound (69). In particular, we prove the following:

**Lemma 10** (Lower bound on testing error). *For the given family of distributions $\{\mathbb{Q}_u, u \in \mathcal{U}\}$, we have*

$$\min_{\substack{u,u' \in \mathcal{U} \\ D_H(u;u')=1}} \inf_\psi \left[ \mathbb{Q}_u(\psi(\mathcal{D}) \neq u) + \mathbb{Q}_{u'}(\psi(\mathcal{D}) \neq u') \right] \geq \left( 1 - \sqrt{\frac{1}{2} \frac{n_H \delta^2}{d^2}} \right). \quad (69)$$

*Thus, the testing error is lower bounded by $\frac{1}{2}$ with the choice $\delta = \frac{d}{\sqrt{2n_H}}$.*

See Section F.4 for the proof.

Combining the claims of Lemmas 9 and 10, along with the choice $\delta = \frac{d}{\sqrt{2n_H}}$, yields the lower bound

$$\sup_{u \in \mathcal{U}} \mathbb{E}_u[V_u^\star - V_u^{\pi_{\mathrm{ALG}}}] \geq \frac{\delta}{\sqrt{2d}} \frac{dH}{2} \frac{1}{2} \geq \frac{1}{8} dH \sqrt{\frac{d}{n_H}}. \quad (70)$$

Thus, the only remaining step is to relate this lower bound to the uncertainty function $\mathcal{U}(\pi; \sqrt{d})$ associated with our family of MDPs. More precisely, we prove the following:

**Lemma 11.** *There is a universal constant such that*

$$\sup_\pi \mathcal{U}(\pi; \sqrt{d}) \leq cdH \sqrt{\frac{d}{n_H}} \quad (71)$$

See Section F.5 for the proof.

Combining Lemma 11 with the lower bound (70) concludes the proof of the theorem.

It remains to prove our auxiliary lemmas, and we do so in the following subsections.

## F.3 Proof of Lemma 9

For a given $u \in \mathcal{U}$, let $\pi_u^\star$ be the optimal policy on $M_u$ and let $V_u^\star$ the optimal value function. For any estimated policy $\pi$, we define the estimated sign vector $u^\pi \in \{-1, 1\}^{dH}$ with entries $[u^\pi]_{hi} \overset{def}{=} \text{sign}(\mathbb{E}_{a \sim \pi_h} a_i)$.

With this set-up, we prove the lemma in two steps:

(a) First, we show that the value function gap $V_u^\star - V_u^\pi$ can be lower bounded in terms of the Hamming distance

$$V_u^\star - V_u^\pi \geq \frac{\delta}{\sqrt{2d}} D_{\mathrm{H}}(u^\pi; u). \tag{72}$$

(b) We use Assaoud's method to lower bound the estimation error in the Hamming distance.

**Step (a):** Since the optimal action at timestep $h$ on $M_u$ is $u_h$, by inspection, the associated suboptimality of $\pi$ on $M_u$ compared to the optimal policy on $M_u$ is

$$
\begin{aligned}
V_u^\star - V_u^\pi &= \frac{1}{\sqrt{2d}} \sum_{h=1}^{H} \left[ \langle u_h, \delta u_h \rangle - \mathbb{E}_{a \sim \pi_h} \langle a, \delta u_h \rangle \right] \\
&= \frac{\delta}{\sqrt{2d}} \sum_{h=1}^{H} \sum_{i=1}^{d} \left[ [u]_{hi}[u]_{hi} - [\mathbb{E}_{a \sim \pi_h} a]_i [u]_{hi} \right] \\
&= \frac{\delta}{\sqrt{2d}} \sum_{h=1}^{H} \sum_{i=1}^{d} \left( [u]_{hi} - [\mathbb{E}_{a \sim \pi_h} a]_i \right) [u]_{hi} \\
&= \frac{\delta}{\sqrt{2d}} \sum_{h=1}^{H} \sum_{i=1}^{d} \left| [u]_{hi} - [\mathbb{E}_{a \sim \pi_h} a]_i \right|.
\end{aligned}
$$

Now recalling that $[u^\pi]_{hi} \overset{def}{=} \text{sign}(\mathbb{E}_{a \sim \pi_h} a_i)$, we have the lower bound

$$
\begin{aligned}
V_u^\star - V_u^\pi &\geq \frac{\delta}{\sqrt{2d}} \sum_{h=1}^{H} \sum_{i=1}^{d} \left| [u]_{hi} - [\mathbb{E}_{a \sim \pi_h} a]_i \right| \mathbb{1}\{u_{hi}^\pi \neq [u]_{hi}\} \\
&\geq \frac{\delta}{\sqrt{2d}} \sum_{h=1}^{H} \sum_{i=1}^{d} \mathbb{1}\{u_{hi}^\pi \neq [u]_{hi}\} \\
&= \frac{\delta}{\sqrt{2d}} D_{\mathrm{H}}(u^\pi; u),
\end{aligned}
$$

which establishes the lower bound (72).

**Step (b):** We can now apply Assouad's method (cf. Lemma 2.12 in the book Tsybakov (2009)), so as to conclude that for any estimated policy $\pi$, we have

$$\sup_{u \in \mathcal{U}} \mathbb{E}_u \left[ D_{\mathrm{H}}(u^\pi; u) \right] \geq \frac{dH}{2} \min_{u, u' | D_{\mathrm{H}}(u; u')=1} \inf_\psi \left[ \mathbb{P}_u(\psi \neq u) + \mathbb{P}_{u'}(\psi \neq u') \right] \tag{73}$$

where $\inf_\psi$ denotes the minimum over all test functions taking values in $\{u, u'\}$.

## F.4 Proof of Lemma 10

We begin by observing that the testing error can be lower bounded in terms of the KL divergence as

$$\min_{\substack{u, u' \in \mathcal{U} \\ D_{\mathrm{H}}(u; u')=1}} \inf_\psi \left[ \mathbb{P}_u(\psi \neq u) + \mathbb{P}_{u'}(\psi \neq u') \right] \geq 1 - \left( \frac{1}{2} \max_{\substack{u, u' \in \mathcal{U} \\ D_{\mathrm{H}}(u; u')=1}} D_{\mathrm{KL}}(\mathbb{Q}_u \| \mathbb{Q}_{u'}) \right)^{1/2}. \tag{74}$$

For instance, see Theorem 2.12 in Tsybakov (2009).

Thus, in order to prove Lemma 10, it remains to bound the Kullback-Leibler divergence of the distributions $\mathbb{Q}_u$ and $\mathbb{Q}_{u'}$ for pairs $u, u' \in \{-1, +1\}^{dH}$ that differ only in a single coordinate.

By construction, the only stochasticity in the dataset lies in the rewards. For any given $u$, equation (67) implies that the distribution over rewards has the product form

$$\mathbb{Q}_u = \prod_{h=1}^{H} \prod_{i=1}^{d} \prod_{j=1}^{\frac{n_h}{d}} \mathcal{N}\left(\frac{e_i^\top}{\sqrt{2d_h}}(\delta u_h), 1\right).$$

Notice that each normal distribution in the above display for $\mathbb{Q}_u$ is identical to the corresponding factor in $\mathbb{Q}_{u'}$ except for the single index in which the vectors $u$ and $u'$ differ. Thus, applying the chain rule for KL divergence yields

$$D_{\mathrm{KL}}(\mathbb{Q}_u \| \mathbb{Q}_{u'}) = \sum_{k=1}^{\frac{n_H}{d}} D_{\mathrm{KL}}(\mathcal{N}\left(\frac{\delta}{\sqrt{2d}}, 1\right) \| \mathcal{N}\left(\frac{-\delta}{\sqrt{2d}}, 1\right)) = \frac{n_H}{2d}\left(2\frac{\delta}{\sqrt{2d}}\right)^2$$

$$= \frac{n_H \delta^2}{d^2},$$

valid for any pair $u, u'$ differing in a single coordinate. Substituting back into the lower bound (74) yields the claim.

### F.5 Proof of Lemma 11

Recall that by definition, we have $\mathcal{U}(\pi; \sqrt{d}) = \sqrt{d}\sum_{h=1}^{H} \|\phi_h^\pi\|_{\Sigma_h^{-1}}$. Consequently, in order to establish the claim, it suffices to show there is a universal constant $c$ such that

$$\sup_{\pi \in \Pi} \|\phi_h^\pi\|_{\Sigma_h^{-1}} \leq c\frac{d}{\sqrt{n_H}} \qquad \text{for each } h \in [H]. \tag{75}$$

Now denote with $[x]_{1:p}$ the first $p$ components of the vector $x$, and with $[x]_p$ the $p$ component of $x$. Using the triangle inequality we can write

$$\|\phi_h^\pi\|_{\Sigma_h^{-1}} \leq \|\left[[\phi_h^\pi]_{1:d}, 0\right]\|_{\Sigma_h^{-1}} + \|\left[0, [\phi_h^\pi]_{d+1}\right]\|_{\Sigma_h^{-1}}.$$

Next, we use a technical lemma to compute the inverse of $\Sigma_h$. By construction $\Sigma_h$ is an arrowhead matrix, i.e., can be written as

$$\Sigma_h = \begin{bmatrix} D & v \\ v^\top & b \end{bmatrix}$$

where we let the normalization constants inside of $\phi$ in Eq. (61) to be

$$\gamma = \frac{1}{\sqrt{2d_h}}, \qquad c = \frac{1}{\sqrt{2}}$$

to define $D \in \mathbb{R}^{d \times d}$ as a diagonal matrix with entries

$$[D]_{ii} = \gamma^2 \frac{n_H}{d} + \lambda$$

and $v \in \mathbb{R}^d$ is a vector with entries

$$[v]_i = \gamma c \frac{n_H}{d}$$

and $b \in \mathbb{R}$ is a scalar

$$b = c^2\left(n_H + \frac{n_H}{d}\right) + \lambda.$$

The inverse of $\Sigma_h$ can then be computed explicitly using known formulas for block matrices or arrowhead matrices. We arrive to

$$\Sigma_h^{-1} = \begin{bmatrix} D' & v' \\ v'^\top & b' \end{bmatrix}$$

where we define the entries in a second. First, the inverse of the Schur complement is

$$b' \stackrel{def}{=} (b - v^\top D^{-1}v)^{-1} = \left( c^2 \left( n_H + \frac{n_H}{d} \right) + \lambda - \sum_{i=1}^{d} \frac{\left( \gamma c \frac{n_H}{d} \right)^2}{\gamma^2 \frac{n_H}{d} + \lambda} \right)^{-1}.$$

Our goal is to show that this is positive, which helps in simplifying the final expression. Notice that

$$\sum_{i=1}^{d} \frac{\left( \gamma c \frac{n_H}{d} \right)^2}{\gamma^2 \frac{n_H}{d} + \lambda} < \sum_{i=1}^{d} \frac{\left( \gamma c \frac{n_H}{d} \right)^2}{\gamma^2 \frac{n_H}{d}} = dc^2 \frac{n_H}{d} = c^2 n_H.$$

Thus

$$(b')^{-1} = \left( c^2 \left( n_H + \frac{n_H}{d} \right) + \lambda - \sum_{i=1}^{d} \frac{\left( \gamma c \frac{n_H}{d} \right)^2}{\gamma^2 \frac{n_H}{d} + \lambda} \right) > c^2 \frac{n_H}{d} + \lambda > 0.$$

These facts imply that the inverse of the above quantity is bounded as

$$b' < \frac{d}{c^2 n_H + d\lambda} < \frac{d}{c^2 n_H}.$$

Continuing the construction of the inverse, we obtain

$$D' = \underbrace{D^{-1}}_{\stackrel{def}{=} D_1'} + \underbrace{D^{-1} v b' v^\top D^{-1}}_{\stackrel{def}{=} D_2'}$$

Noice that $D_1'$ is symmetric positive definite with positive diagonal elements and $D_2'$ is also symmetric positive semidefinite:

$$0 \prec D_1' = D^{-1} = \left( \gamma^2 \frac{n_H}{d} + \lambda \right)^{-1} I \prec \frac{d}{\gamma^2 n_H} I$$

$$D_2' = \underbrace{b'}_{\geq 0} \underbrace{D^{-1}v}_{y} \underbrace{v^\top D^{-1}}_{y^\top} = b' y y^\top \succcurlyeq 0.$$

We now use the above block expressions for $\Sigma_h^{-1}$ to bound

$$\|\phi_h^\pi\|_{\Sigma_h^{-1}} \leq \| \left[ [\phi_h^\pi]_{1:d}, 0 \right] \|_{\Sigma_h^{-1}} + \| \left[ \vec{0}, [\phi_h^\pi]_{d+1} \right] \|_{\Sigma_h^{-1}}.$$

By construction, $[\phi_h^\pi]_{1:d}$ only interacts with the $D'$ block in $\Sigma_h^{-1}$; using this and

$$\|x\|_{D'}^2 = x^\top (D_1' + D_2')x \leq \|x\|_2 \left( \|D_1'\|_2 + \|D_2'\|_2 \right) \|x\|_2$$

we can write

$$\| \left[ [\phi_h^\pi]_{1:d}, 0 \right] \|_{\Sigma_h^{-1}} = \|[\phi_h^\pi]_{1:d}\|_{D'} \leq \|[\phi_h^\pi]_{1:d}\|_2 \sqrt{\|D_1'\|_2 + \|D_2'\|_2}$$

Likewise,

$$\| \left[ 0, [\phi_h^\pi]_{d+1} \right] \|_{\Sigma_h^{-1}} = \|[\phi_h^\pi]_{d+1}\|_{b'}.$$

We now bound all norms:

$$\|[\phi_h^\pi]_{1:d}\|_2 \leq \frac{\|\mathbb{1}\|_2}{\sqrt{2d}} \leq \frac{1}{\sqrt{2}}$$

$$\|D_1'\|_2 = \|D^{-1}\|_2 \lesssim \frac{2d^2}{n_H}$$

$$\|D_2'\|_2 \leq b'\|D^{-1}\|_2 \|v\|_2 \|v\|_2 \|D^{-1}\|_2 \lesssim \underbrace{\frac{d}{n_H}}_{b'} \underbrace{\left( \gamma \frac{n_H}{d} \right)^2 \|\mathbb{1}\|_2^2}_{\|v\|_2^2} \underbrace{\frac{d^4}{n_h^2}}_{\|D^{-1}\|_2^2} \lesssim \frac{d^2}{n_H}$$

Substituting back yields the bound

$$\| \left[ [\phi_h^\pi]_{1:d}, 0 \right] \|_{\Sigma_h^{-1}} \lesssim \frac{d}{\sqrt{n_H}}$$

Similarly, we have

$$\|[\phi_h^\pi]_{d+1}\|_{b'} = \sqrt{\frac{1}{\sqrt{2}} b' \frac{1}{\sqrt{2}}} \leq \sqrt{\frac{1}{2} \frac{d}{c^2 n_H}} \lesssim \frac{\sqrt{d}}{\sqrt{n_H}}.$$

Putting together the pieces yields the claim (75).