# OpenReview forum: "Provable Benefits of Actor-Critic Methods for Offline Reinforcement Learning"
_NeurIPS.cc/2021/Conference — NeurIPS 2021 Poster_

### Official Review · Reviewer_3ZfY · 2021-07-16

**Rating:** 6
**Confidence:** 1

**Summary:**

The authors propose a new actor-critic method for offline RL which is based on pessimism through perturbations applied to the weights. The authors prove a lower bound on the performance of the policy, while noting their approach requires fewer assumptions on function class and MDP, as well as computational cost, than previous methods with similar theoretical guarantees.

**Limitations And Societal Impact:**

I believe the authors have addressed the limitations adequately.

**Main Review:**

I'm not particularly knowledgeable about the theoretical aspects of this area. I'm going to comment on the practical implications of this work regardless, but my impression is that this work doesn't need to have immediate empirical benefits to provide value to authors working on related topics. My confidence score reflects my level of expertise in this area and I recommend the AC puts little emphasis on my final score.

Are there practical benefits/observations/analysis for somebody who's objective is performance (i.e. leveraging state-of-the-art deep RL methods), in this paper? Probably not.
- The obvious reason is the limitations due to necessary assumptions made by the authors + the lack of empirical results, as pointed out by the authors themselves, means the practical benefits of the proposed algorithm are difficult to evaluate or extend to existing algorithms.
- One claim the authors make is that the paper analyzes actor-critic methods in the context of offline RL. This claim isn't really reflected in the content of the paper, which is more about the proposed algorithm, rather than an analysis on actor-critic methods as a whole. It doesn't seem like we could take this analysis and extend it to existing actor-critic approaches such as BCQ [1] or CQL [2].
- I think an underappreciated limitation is on the data generation process. The practical side of offline RL really centers around the use of a finite-sized dataset.
- To the best of my knowledge, the use of parameter perturbation to induce pessimism for offline RL is novel in context of the deep RL, but has been used previously for exploration [3,4]. And the connection between pessimism and exploration has been noted (i.e. [1,5,6]), so I wouldn't consider this a significant algorithmic novelty.

Comments on clarity:
- I think the paper would benefit from more guiding statements. For example, section 3 jumps immediately into algorithm description, without providing any overview. This also applies to section 4.

Comments on prior work:
- The author mentions pessimism as a common approach in offline RL, but in the deep RL literature, policy constraint approaches are vastly more popular (such as [1,7,8,9]).

I'll re-iterate that much what I've discussed should not necessarily be considered relevant, but I'll leave it to the AC to determine what should be considered. I've left my score as positive as I believe the authors have achieved what they have set out to do.

References
- [1] Fujimoto, Scott, David Meger, and Doina Precup. "Off-policy deep reinforcement learning without exploration." International Conference on Machine Learning. PMLR, 2019.
- [2] Kumar, Aviral, et al. "Conservative q-learning for offline reinforcement learning." arXiv preprint arXiv:2006.04779 (2020).
- [3] Fortunato, Meire, et al. "Noisy Networks For Exploration." International Conference on Learning Representations. 2018.
- [4] Plappert, Matthias, et al. "Parameter Space Noise for Exploration." International Conference on Learning Representations. 2018.
- [5] Laroche, Romain, Paul Trichelair, and Remi Tachet Des Combes. "Safe policy improvement with baseline bootstrapping." International Conference on Machine Learning. PMLR, 2019.
- [6] Buckman, Jacob, Carles Gelada, and Marc G. Bellemare. "The Importance of Pessimism in Fixed-Dataset Policy Optimization." International Conference on Learning Representations. 2020.
- [7] Kumar, Aviral, et al. "Stabilizing Off-Policy Q-Learning via Bootstrapping Error Reduction." Advances in Neural Information Processing Systems 32 (2019): 11784-11794.
- [8] Wu, Yifan, George Tucker, and Ofir Nachum. "Behavior regularized offline reinforcement learning." arXiv preprint arXiv:1911.11361 (2019).
- [9] Kostrikov, Ilya, et al. "Offline reinforcement learning with fisher divergence critic regularization." International Conference on Machine Learning. PMLR, 2021.

**Time Spent Reviewing:**

5

---

> ### Author Response · Authors · 2021-08-10
> **Connections with the Deep RL literature and foundational results**
>
> We thank the reviewer for connecting this work with the broader deep RL literature.
>
> A goal we had with this work was to see whether existing theoretical analyses with provable guarantees with function approximation (which are very recent and mostly for value-based methods) could be improved by means of a different class of algorithms.
>
> We found an actor-critic formulation to be quite natural to solve the max-min optimization problem, and we then focused on designing a specific algorithm and setting where we could precisely lay out the assumptions (in terms of function class), the statistical rates and distance to optimality to clearly show the benefits. While these advantages are indeed enabled by an actor-critic approach, we completely agree with the reviewer that this should not be taken as a more general claim about actor-critic methods. As the reviewer notes, parameter space perturbation has been previously considered; here we consider a specific form of perturbation to achieve tight performance bounds for the setting we consider. We will make sure to clarify the last two points in the text.
>
> We acknowledge that often theoretical studies trail behind what’s done by practitioners with more complex models. Despite our findings likely won’t have an immediate impact to the design of deep RL algorithms, our hope is that they serve as a first solid attempt to better understand the capabilities, the limitations and the trade-offs of RL algorithms when simple (= linear) function approximation schemes are implemented.

---

### Official Review · Reviewer_Vb58 · 2021-07-18

**Rating:** 8
**Confidence:** 3

**Summary:**

The paper presents a theoretical study on -- quote – “Do actor-critic methods provably offer any advantage in offline RL?” This is a fundamental question to offline RL, and the authors give a positive answer to that question. Compared to Jin et al. 2020b which uses value-based RL method (PVEI), the main result of the paper is i) a tighter gap between the lower and upper bounds of the value functions obtained by their proposed Pessimistic Least Square Policy Evaluation method and ii) relaxed assumptions about the MDP (restricted closeness vs linear MDP). Actor critic RL is a predominant method in RL, and thus the new findings in this paper are really impressive and encouraging.

**Limitations And Societal Impact:**

A limitation in my opinion is the assumption of functional spaces which is a key ingredient in guaranteeing the performance bound. However, I think the assumption is still reasonable by large. I understand that it is challenging to obtain performance guarantee without such assumptions.

**Main Review:**

The paper is very well written in general, with a nice layout beginning with explicit definition of the problem setting and line of assumptions, ending with main result and its implications, as well as the key ingredients of proof.
Detailed comments:
1)	The performance guarantee of actor critic methods in offline RL is a fundamental problem.
2)	The main results are positive and encouraging.
3)	The implications of the results are well discussed.
4)	The pessimistic perturbation of the critic instead of using additive bonuses or absorbing states, so that by the carefully constructed value and policy function classes, the linearity of the action value function is maintained, which is critical to the proof.
5)	The line of research has been comprehensively discussed, and the new contribution in this paper has been clearly pointed out and well highlighted. E.g., the comparison against Jin et al. 2020b clearly shows the new advantage of actor critic methods over pessimistic value iteration methods.

A limitation in my opinion is the assumption of functional spaces which is a key ingredient in guaranteeing the performance bound. However, I think the assumption is still reasonable by large.
A minor point – a typo in Line 157: “cf. Line 1”-> “cf. Line 5”


**Time Spent Reviewing:**

7 hours

---

> ### Author Response · Authors · 2021-08-10
> **Trade-off between assumptions and computational tractability**
>
> We thank the reviewer for the positive review.
>
> Indeed, in light of recent lower bounds for offline RL, some assumptions beyond realizability of the predictor are needed to obtain meaningful guarantees. Our assumption sits somewhere in the middle between the linear Q^\pi setting where sample efficient batch RL is not possible (e.g., Zanette 2021, Wang et al 2021) and the well studied low-rank model of Jin et al, 2020 where statistical and computational tractability can be easily achieved.
>
> With this in mind, with this work we wanted to strike a balance between the generality of the formulation (= ‘weaker assumptions’) and other competing goals like computational tractability: while the assumption that we make on the function class are more general than the low rank model, they still allow us to design a computationally tractable algorithm which we think is very important in a path towards possible applications.

---

> > ### Comment · Reviewer_Vb58 · 2021-09-01
> > **Thank you for the response**
> >
> > Good work -- I enjoyed reading the paper!

---

### Official Review · Reviewer_rEwm · 2021-07-24

**Rating:** 8
**Confidence:** 4

**Summary:**

This paper presents actor-critic methods for offline RL. Specific aspects of the problem addressed by the paper include being able to achieve pessimism without adding in pessimism based bonuses or through defining absorbing states. This allows the approach to work with the original value function class without modifications. The paper also presents distinctions between standard assumptions used in both online/offline RL literature relating to low rank MDPs, restricted closeness, and linear Q^\pi assumption.

**Limitations And Societal Impact:**

Yes.

**Main Review:**

The result presented by this paper appears to be novel, and the algorithm is computationally efficient. What I like about the main result is that  it tends to obtain a result that is competitive with all policies (beyond ones in the policy class considered), such that modulo optimization error, the algorithm finds a policy competitive with the one with the best value subtracted by some form of uncertainty function defined according to the dataset. This result is complemented by an upper-bound that shows that this cannot really be improved upon beyond some constant factors and log factors.

Minor comments:
- note that in the work of Kidambi et al. 2020, the comparator policy can be an arbitrary policy (not necessarily the optimal policy), albeit, this paper's result is stronger for reasons mentioned above. Could the authors verify if this is the case with the other works they mention below Theorem 2?

**Time Spent Reviewing:**

3-4

---

> ### Author Response · Authors · 2021-08-10
> **Comparison with the literature regarding arbitrary comparator policies**
>
> We thank the reviewer for the positive review. Here we address the point raised by the reviewer.
>
> ``Minor comments: note that in the work of Kidambi et al. 2020, the
> comparator policy can be an arbitrary policy (not necessarily the
> optimal policy), albeit, this paper’s result is stronger for reasons
> mentioned above. Could the authors verify if this is the case with the
> other works they mention below Theorem 2?’’
>
> Thanks for catching this: indeed Kidambi et al can similarly
> (like in Liu et al and Yu et al that we cite) compare to arbitrary
> policies. To address your question, yes, the other prior work we cite
> below Thm 2 (Jin et al. 2020b; Kumar et al. 2019; Buckman et al 2020)
> compare to \pi*. However, the key benefit of our analysis is to
> significantly improve beyond the prior results in both sets of papers
> in terms of the tightness under less restrictive assumptions than
> prior work (lines ~214-221). We will clarify this in the text.

---

### Official Review · Reviewer_Zu4e · 2021-08-02

**Rating:** 6
**Confidence:** 4

**Summary:**

This paper investigates whether the actor-critic methods have probable advantages compared to the model or value-based approaches in the offline reinforcement learning (RL) setting. Recent work on offline RL propose to incorporate pessimism into the procedure and this work argues that the actor-critic methods can naturally achieve so by separating the policy optimization from the policy evaluation.

**Limitations And Societal Impact:**

There are a few limitations in the current version. In particular, it is not clear whether designing the goal as finding the policy with highest minimum value function is the best choice (or a better choice than alternative choices) in the offline RL setting. Actually, while it is desirable to introduce pessimism into the framework in face of uncertainty, the design of the goal itself in this work could be too restrictive.

**Main Review:**

Overall, this work chooses an important problem to work on and have shown certain advantages of actor-critic methods in the offline RL setting. There is a re-surging interest in offline RL. While most recent work focues on the model or value-based methods, this work instead explore the popular actor-critic methods given only logged data, which may motivate further investigation along this line.

**Time Spent Reviewing:**

2

---

> ### Author Response · Authors · 2021-08-10
> **Finding the policy with highest minimum value function is a sensible objective for risk-adverse applications**
>
> We thank the reviewer for the positive feedback. We agree that different applications of offline RL naturally lead to different objectives: in certain cases we may be satisfied with returning a policy with high expected value, and in some other cases strong prior knowledge may call for a bayesian approach.
>
> Our work builds on the recent literature whose objective is to return with high probability a policy with guaranteed value as high as possible; in line with the past literature, we think this continues to be a sensible objective for risk-adverse offline RL, despite it is not the only possible one, as recognized by the reviewer.

---

### Decision · Program_Chairs · 2021-09-27

**Decision:**

Accept (Poster)

**Comment:**

This paper proposes an actor-critic style algorithm for solving the offline RL problem. The key contribution is in showing that pessimism can be naturally incorporated in this framework, designing an efficient algorithm with provable performance guarantees. At the same time, the reviewers also point out limitations/points of discussion such as whether the goal of finding the policy with highest minimum value function is the best choice (or a better choice than alternative choices) in the offline RL setting, discussion of assumptions on functional space and data generation process -- having a discussion around these and other points in the reviews would improve the quality of paper.

Overall, since all reviewers agree that this paper makes a good theoretical contribution to the offline RL problem, I am recommending acceptance.